# Structural insights into the ubiquitylation strategy of the oligomeric CRL2[FEM1B] E3 ubiquitin ligase

Zonglin Dai [ID] [1,2,8], Ling Liang [ID] [1,3,8], Weize Wang [1,4,8], Peng Zuo [1,2], Shang Yu [ID] [3], Yaqi Liu [5], Xuyang Zhao [1], Yishuo Lu [ID] [1,4], Yan Jin [1], Fangting Zhang [6], Dian Ding [ID] [1], Weiwei Deng [7] & Yuxin Yin [ID] [1,2,4,6✉]

## Abstract

**Cullin-RING E3 ubiquitin ligase (CRL) family members play critical roles in numerous biological processes and diseases including cancer and Alzheimer's disease. Oligomerization of CRLs has been reported to be crucial for the regulation of their activities. However, the structural basis for its regulation and mechanism of its oligomerization are not fully known. Here, we present cryo-EM structures of oligomeric CRL2[FEM1B] in its unneddylated state, neddylated state in complex with BEX2 as well as neddylated state in complex with FNIP1/FLCN. These structures reveal that asymmetric dimerization of N8-CRL2[FEM1B] is critical for the ubiquitylation of BEX2 while FNIP1/FLCN is ubiquitylated by monomeric CRL2[FEM1B]. Our data present an example of the asymmetric homo-dimerization of CRL. Taken together, this study sheds light on the ubiquitylation strategy of oligomeric CRL2[FEM1B] according to substrates with different scales.**

**Keywords** Cryo-EM; Cullin-RING E3 Ubiquitin Ligase; Ubiquitylation; Oligomerization
**Subject Categories** Post-translational Modifications & Proteolysis; Structural Biology

## Introduction

Cullin-RING E3 ubiquitin ligases (CRLs) are multi-subunit complexes composed of a catalytic Ring-finger protein, a cullin scaffold protein, substrate adapter proteins, and a substrate recognition protein (Sarikas et al, 2011). CRLs have crucial roles in controlling the cell cycle (Jang et al, 2020), hypoxia signaling (Maxwell et al, 1999), reactive oxygen species clearance (Kensler et al, 2007), and DNA repair (Ferretti et al, 2016), all of which are pivotal processes regulating several disorders including cancer

(Zhao and Sun, 2013), Alzheimer's disease (Potjewyd and Axtman, 2021), and delayed development (Jiang et al, 2012), as well as tissue response to ionizing radiation (Fouad et al, 2019). In the ubiquitin-proteasome system (UPS), CRLs act as a platform for polyubiquitin chain formation where E2 ubiquitin-conjugating enzymes (E2s) are recruited to the catalytic domain and substrate binds to the substrate recognition domain (Baek et al, 2021; Nguyen et al, 2017). Ubiquitin activating enzyme (E1) chemically activate a ubiquitin molecule in an ATP-dependent manner and transfer it to E2s through a *trans*-thiolation reaction. Ultimately, the CRLs promote the ubiquitin modification of lysine residues on their substrates (Kleiger and Mayor, 2014; Liu et al, 2020). However, the catalytic domain on the C terminal domain of cullin scaffold is around 10 nm away from the substrate recognition domain on the N terminal domain of cullin scaffold (Zhou et al, 2021). Therefore, investigating the ubiquitylation strategies of CRLs in relation to substrates of varying sizes will enhance our understanding of the ubiquitylation process.

From the perspective of protein evolution, it has previously been shown that protein oligomerization offers new opportunities for functional control and higher order complexity (Ali and Imperiali, 2005; Hashimoto and Panchenko, 2010; Kumari and Yadav, 2019). In the past 15 years, several studies have shown that oligomerization of CRLs plays critical roles in activity regulation of ubiquitin transfer (Bulatov and Ciulli, 2015; Bulatov et al, 2015; Zimmerman et al, 2010). For example, studies of neddylated CRL1[FBXW7] and ARIH1 hetero-dimeric E3-E3 super-assemblies revealed how two types of E3s co-evolved to transfer ubiquitin to various substrates with folded structures or of limited lengths (Horn-Ghetko et al, 2021). It has also been reported that CUL2 assembles hetero-dimeric CRLs with CUL4A (which is involved in the progression of Alzheimer's disease) (Yasukawa et al, 2020), in addition to forming a homo-dimeric CRL2[VHL] complex in vivo (Chung et al, 2006; Merlet et al, 2009). Although homo-dimeric Von Hippel Lindau protein (pVHL) could not be detected in vitro, these studies provide strong evidence that oligomerization of CRLs occurs not only via substrate recognition proteins but also via adapters, Cullin scaffolds, or their combinations (Bulatov and Ciulli, 2015).

[1]Institute of Systems Biomedicine, Beijing Key Laboratory of Tumor Systems Biology, School of Basic Medical Sciences, Peking University Health Science Center, Beijing 100191, China. [2]Department of Pathology, School of Basic Medical Sciences, Peking University Health Science Center, Beijing 100191, China. [3]Department of Biophysics, School of Basic Medical Sciences, Peking University Health Science Center, Beijing 100191, China. [4]Peking-Tsinghua Center for Life Sciences, Peking University, Beijing 100871, China. [5]Department of Physiology and Cellular Biophysics, Clyde and Helen Wu Center for Molecular Cardiology, Department of Medicine, Columbia University Vagelos College of Physicians and Surgeons, New York, NY 10032, USA. [6]Institute of Precision Medicine, Peking University Shenzhen Hospital, Shenzhen 518036, China. [7]Department of Mechanics and Aerospace Engineering, Southern University of Science and Technology, Shenzhen 518055, China. [8]These authors contributed equally: Zonglin Dai, Ling Liang, Weize Wang. ✉E-mail: yinyuxin@hsc.pku.edu.cn

However, the physiological role of oligomerization of CRL2s is not yet well understood.

FEM1B is the substrate recognition protein of the oligomeric CRL2$^{FEM1B}$ E3 ubiquitin ligase (Dankert et al, 2017; Koren et al, 2018; Lin et al, 2018; Wang et al, 2016) that has been implicated in sex determination (Starostina et al, 2007), apoptosis (Chan et al, 2000), colon cancer (Subauste et al, 2010) and Alzheimer's disease (Crist et al, 2021). Several studies have recently focused on the various recognition mechanisms of different substrates by FEM1B. CRL2$^{FEM1B}$ regulates reductive stress via the proteasomal degradation of FNIP1/FLCN complex in a Zn$^{2+}$-dependent manner. This process could be inhibited by BEX proteins, which function as competitive inhibitors (Manford et al, 2021; Manford et al, 2020). The finding that a BEX-binding deficient mutant FEM1B$^{R126Q}$ is associated with syndromic global developmental delay (Lecoquierre et al, 2019) highlights the importance of the regulation of CRL2$^{FEM1B}$ by BEX proteins. However, whether CUL2$^{FEM1B}$ can ubiquitylate BEX proteins as well as the involved regulatory mechanism remain to be determined. It has also been proposed that CRL2$^{FEM1B}$ might be involved in neurologic disorders, such as Alzheimer's, by regulating the function of cyclin-dependent kinase 5 activator 1 (CDK5R1) (Chow et al, 2019; Draney et al, 2016; Liu et al, 2019; Moncini et al, 2016; Spreafico et al, 2018; Zeng et al, 2021). As evidenced by crystal structures, FEM1B selectively binds to the Arg/C-degron of CDK5R1 (Chen et al, 2021; Yan et al, 2021).

Although the FEM1B residues critical for the recognition of Arg/C-degron of CDK5R1 differ from those required for FNIP1 or BEX proteins, FEM1B engages all these targets in a similar deep groove on its concave side, localized to the N-terminal ankyrin repeats and tetratricopeptide repeat (TPR) motif (Chen et al, 2021; Manford et al, 2021). Although these structural studies of the N-terminal domain of FEM1B and peptide substrates provided critical insights into the substrate recognition mechanisms of CRL2$^{FEM1B}$, a comprehensive mechanistic understanding of the substrate recognition of oligomeric full-length CRL2$^{FEM1B}$ is still to be determined.

Here, we describe for the first time the cryo-EM structures of oligomeric CRL2$^{FEM1B}$ in three different states as follows: unneddylated CRL2$^{FEM1B}$, dimeric neddylated CRL2$^{FEM1B}$ complexed with BEX2, and monomeric neddylated CRL2$^{FEM1B}$ complexed with FNIP1/FLCN. Unexpectedly, we found that dimeric neddylated CRL2$^{FEM1B}$ complexed with BEX2 adopted similar conformations as dimeric unneddylated CRL2$^{FEM1B}$. Our in vitro molecular biology experiments showed that dimeric N8-CRL2$^{FEM1B}$ is crucial for the ubiquitylation of BEX2, while the ubiquitylation of FNIP1/FLCN is facilitated by monomeric N8-CRL2$^{FEM1B}$. It suggested that oligomerization of CRL2$^{FEM1B}$ provided the structural basis for ubiquitylation of different substrates. Taken together, our results provide a different perspective to the substrate-ubiquitylation strategy of oligomeric CRL2$^{FEM1B}$ complex and suggest that RBX1 can take part not only in the E2~Ub binding but also oligomerization of CRLs.

## Results

### Cryo-EM structure of CRL2$^{FEM1B}$ complex shows asymmetric super-assembly

CRL2$^{FEM1B}$ is an oligomeric E3 ubiquitin ligase composed of CUL2, FEM1B, ELOB, ELOC, and RBX1. To gain molecular insights into

the assembly of CRL2$^{FEM1B}$, we recombinantly expressed and purified CRL2$^{FEM1B}$ by IMAC and anion exchange, and then used cryo-EM to solve its structure (Appendix Fig. S1A–D). Our initial recombinant CRL2$^{FEM1B}$ preparation showed considerable heterogeneity as shown in Appendix Fig. S1E; greater homogeneity was achieved using GraFix (gradient fixation), which substantially reduced the background of fragmented particles (Appendix Fig. S1F). Using conventional single particle analysis cryo-EM method, we solved the structure of purified recombinant CRL2$^{FEM1B}$ to 4.08 Å resolution (Appendix Table S1 and Appendix Fig. S2A,B). The asymmetric super-complex is shown in Fig. 1A,B. The asymmetric super-complex consists of three protomers, but only protomer 1 and protomer 2 were well resolved. Protomer 3 was partially obscured and requires a lower contour threshold level to be visible (Fig. 1B). The high-resolution cryo-EM map enabled us to dock all the available crystal structures of CUL2, RBX1, ELOB, ELOC, and FEM1B (1-337) (PDB 5N4W, 6LBF) with a high degree of confidence and adjust the model according to the densities of the components (Fig. 1C–E). For the unresolved C-terminus of FEM1B (residues 338–627), we built a de novo model based on the densities of the bulky side chains of the Trp, Phe, and Tyr residues (Appendix Fig. S2C).

### Organizational architecture of CRL2$^{FEM1B}$

A single protomer of CRL2$^{FEM1B}$ in the oligomers has a conformation similar to CRL2$^{VHL}$ (Cardote et al, 2017) with the notable exception that FEM1B adopts a different 'elbow-like' fold. The superposition of the two structures reveals a slight difference in the CUL2-RBX1 scaffolds and adapter proteins, with a Cα root-mean-square deviation of 2.088 Å. (Appendix Fig. S6A). Moreover, the loop connecting the WHB domain to the rest of the C-terminus of CUL2 is not visible in either of the structures, possibly due to its flexibility, which may facilitate conformational changes of the CUL2 WHB domain (Appendix Fig. S6A,B).

FEM1B is organized with six annotated ankyrin repeats in the N-terminal domain and two ankyrin repeats in the C-terminal domain, bridged by one TPR repeat and two α-helical regions (Fig. 2A). The crescent-shaped N-terminal ankyrin repeats as well as the α-helical region and TPR repeat are consistent with recent reports by others (Chen et al, 2021; Yan et al, 2021) (Appendix Fig. S6D). In the C-terminal domain, the TPR repeat is followed by another α-helical region and two ankyrin repeats (Fig. 2A). The TPR repeat forms the turning point of the 'elbow' with the N-terminal domain and C-terminal domain elongating on each side of it. The groove on the concave side of the N-terminal domain provides a deep binding pocket for substrates (Manford et al, 2021). To accommodate substrates of different sizes, FEM1B is a highly dynamic protein in which the movement of the TPR repeat alters the angle of the 'elbow' and thus the size of the binding pocket (Appendix Fig. S10A).

In the C-terminus of FEM1B, the canonical VHL box interacts with ELOC and CUL2 in a manner similar to that reported for VHL (Cardote et al, 2017) (Fig. 2B; Appendix Fig. S6C). The VHL box is comprised of two components with a BC box (residues 597–608) mediating the association with ELOC and a cullin box (residues 617–627) interacting with CUL2 (Fig. 2B–D). The loop connecting the BC box to the cullin box in FEM1B is shorter than the comparable loop in VHL (Fig. 2B; Appendix Fig. S6C).

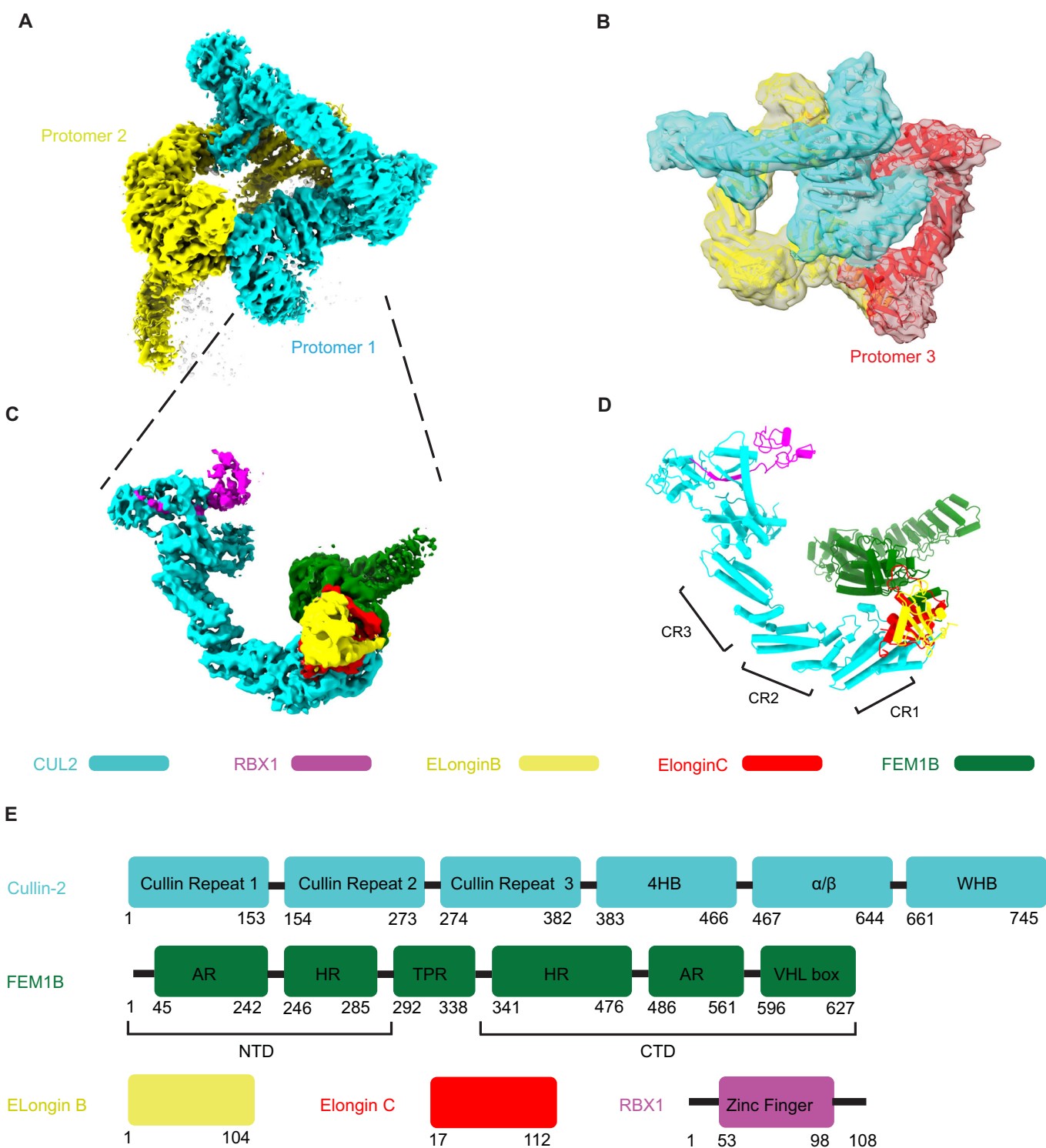

**Figure 1.   The CRL2^FEM1B assembles as asymmetric super-complex.**

(**A**) Near atomic-resolution cryo-EM map of dimeric CRL2^FEM1B with protomer 1 in cyan and protomer 2 in yellow. (**B**) Overview of the low-pass filtered cryo-EM map of CRL2^FEM1B with protomer 1 cyan, protomer 2 yellow, and protomer 3 red. (**C,D**) Cryo-EM map of protomer 1 (**C**) with fitted atomic models of CUL2 (cyan), RBX1 (magenta), ELOB (yellow), ELOC (red), and FEM1B (green) shown in cylinder representation (**D**). (**E**) Schematic representation of the components of CRL2^FEM1B and the constructs used for cryo-EM with each of the subunits assigned a different color the same as in panel (**C**).

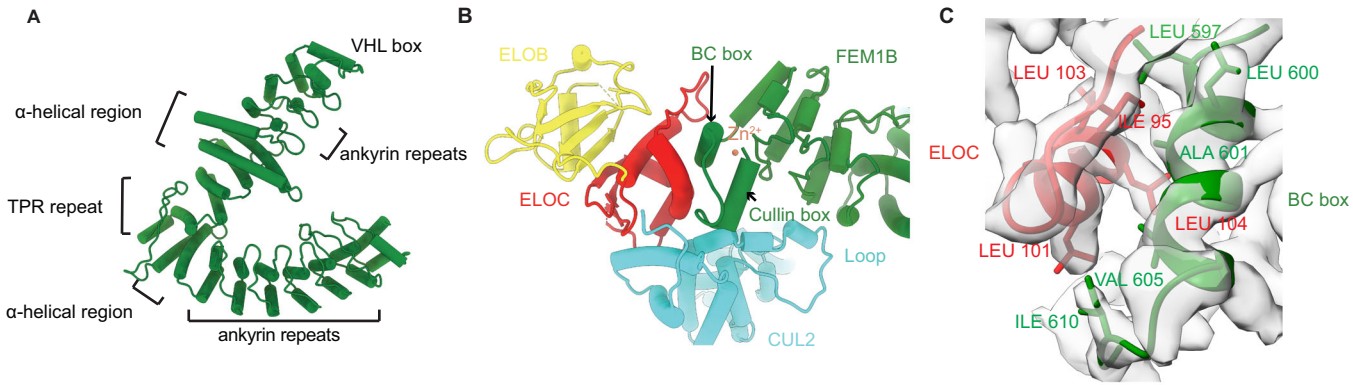

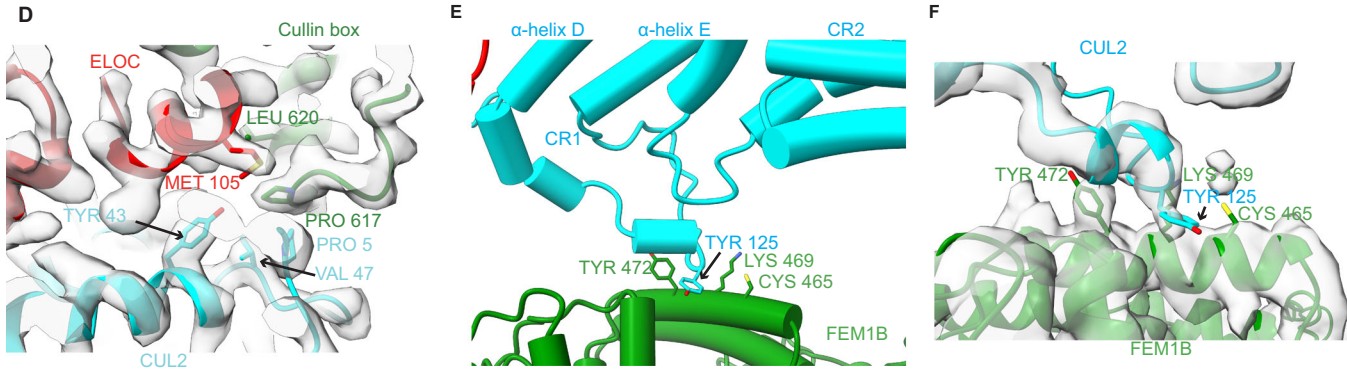

**Figure 2. Interactions within the CRL2^FEM1B complex.**

(**A**) Domain structure of FEM1B which contains six ankyrin repeats at its N-terminus, two ankyrin repeats and a VHL box in its C-terminal region, and the intervening TPR domain flanked on both sides by an α-helical region. (**B**) Overview of the interactions among FEM1B, ELOC, and CUL2. (**C**) Details of the atomic interactions between the FEM1B BC box and ELOC. (**D**) Three-way atomic interactions among the FEM1B Cullin box, ELOC, and CUL2. (**E,F**) Details of the atomic interactions between FEM1B and the loop connecting α-helix D and α-helix E of CUL2 Cullin Repeat 1 (CR1).

Furthermore, the loop connecting α-helix D and α-helix E of the CUL2 cullin repeat 1 (which was not resolved in the previously reported X-ray crystal structure (Cardote et al, 2017)), appears to bind the C-terminus of FEM1B through hydrophobic interactions mediated mainly by Tyr125 of CUL2 and Cβ-Cε of Lys469, Tyr472, and Cys521 of FEM1B. These interactions in turn might further stabilize CRL2^FEM1B (Fig. 2E,F). Taken together, our data indicate that FEM1B is a VHL box containing substrate recognition protein with distinctive properties that suggest it is stabilized by a unique loop of CUL2.

## Mechanism of CRL2^FEM1B supercomplex assembly

Protomer 1 within the homo-dimeric CRL2^FEM1B super-complex acts as a scaffold stabilizing this auto-inhibited super-assembly (Fig. 3A). In addition, extensive hydrophobic interactions are formed between the N-terminal domain α-helix of FEM1B (residues 269–284) in protomer 1, and the C-terminal domain ankyrin repeats (residues 546–588) of FEM1B in protomer 2 (Fig. 3A,B). These interaction regions are unique for FEM1B among FEM1 family members (Appendix Fig. S7A) and are less conserved

during evolution (Appendix Fig. S7B), which may explain why FEM1B oligomerizes whereas FEM1A and FEM1C do not (Chan et al, 1999; Consortium, 2021). The CUL2-RBX1 scaffold is also involved in the interaction between protomer 1 and protomer 2. Thus, the C-terminal WHB domain of CUL2 (661–728) and RBX1 in protomer 1 form a 'forceps'-like conformation, 'clamping' the cullin repeat 3 of CUL2 in protomer 2 (Fig. 3C,D). In this way, the 'head' and 'body' of protomer 2 are completely fixed by protomer 1. Meanwhile, the α-helix repeats in the FEM1B N-terminus in protomer 1 bind to the ankyrin repeats in the FEM1B C-terminal domain in protomer 3 (Fig. 3E). Furthermore, the TPR domain of protomer 2 and protomer 3 are in contact with each other (Fig. 3F).

To test whether perturbation of the interface can affect its oligomerization, we expressed and purified mutants FEM1B^F549R, FEM1B^F549S, FEM1B^F549T, FEM1B^Y275R/L278R, and FEM1B^Δ546-553 (FEM1B^DEL) based on our cryo-EM structures. NanoDSF measurement results suggested that these mutations had no significant influence on the protein stabilities (Table 1; Appendix Fig. S11). Mass photometry (MP) experiments were performed to investigate the oligomerization behavior of wild-type CRL2^FEM1B and its oligomerization-deficient mutants described above. As we expected,

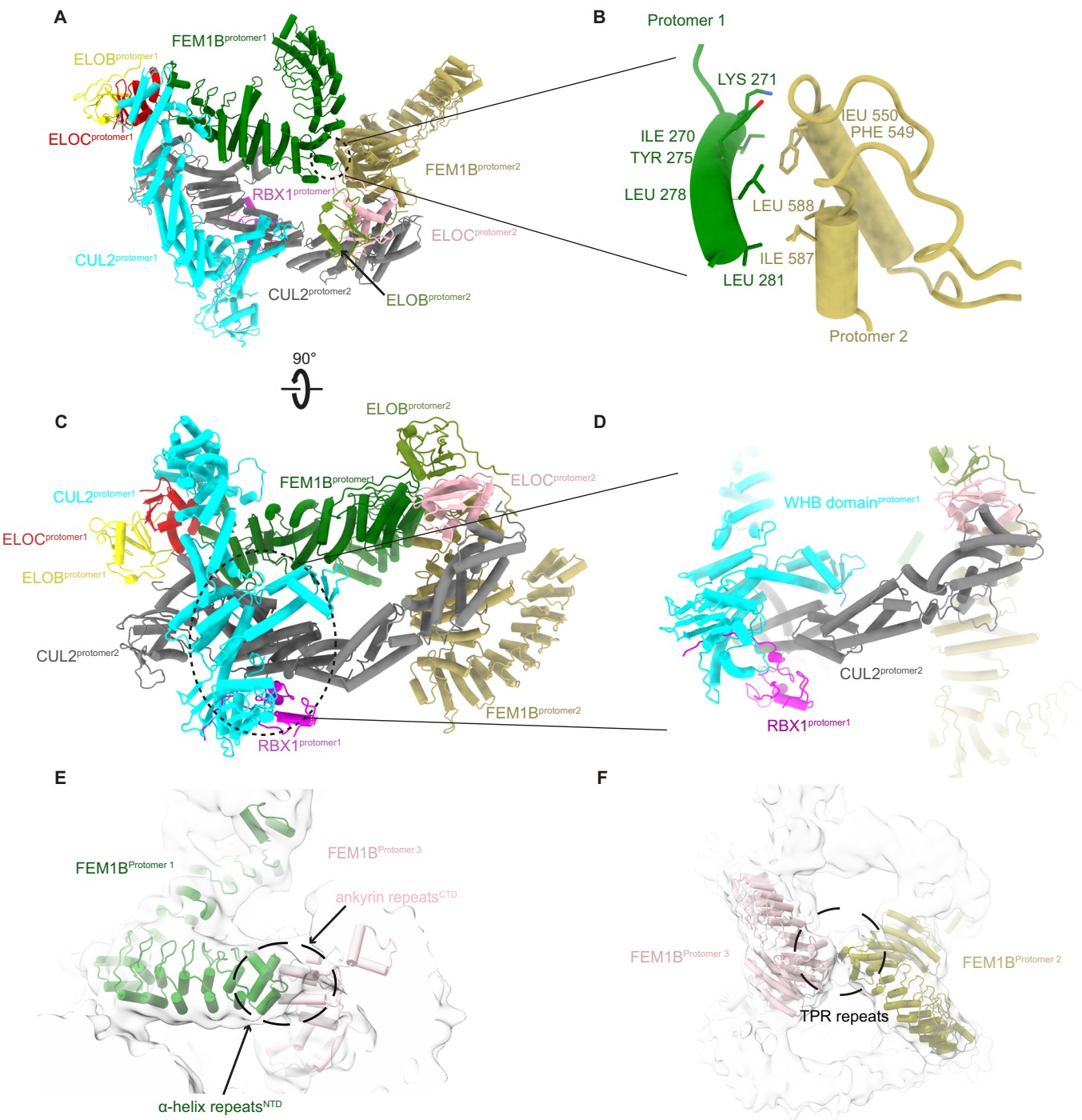

**Figure 3. Architectural organization of the CRL2^FEM1B supercomplex.**

(**A**) Overview of the interactions between protomer 1 and protomer 2 of FEM1B. (**B**) Expanded view of (**A**) showing details of the hydrophobic interactions between the N-terminal α-helix of FEM1B (residues 269–284) in protomer 1 and the C-terminal ankyrin repeats of FEM1B (residues 546–588) in protomer 2. (**C,D**) Two views showing how the cullin repeat 3 of CUL2 in protomer 2 is clamped by a 'forceps' conformation formed by RBX1 and the WHB domain of CUL2 in protomer 1. (**E**) Low-pass filtered cryo-EM map fitted with an atomic model of the protomers that shows the interactions between the N-terminal domain of protomer 1 (green) and the ankyrin repeats in the C-terminus of protomer 3 (pink). (**F**) Atomic models fitted in the low-pass filtered cryo-EM map show that the TPR repeats of FEM1B of protomer 2 and protomer 3 are involved in the formation of trimeric CRL2^FEM1B. Circled region corresponds to the contacted TPR repeats of both protomers.

**Table 1.** The effect of mutations on CRL2FEM1B and MBP-FCB protein stability ($n = 3$).

| Samples | Tm (°C) |
|---|---|
| CRL2FEM1B-WT | 56.28 ± 0.07 |
| CRL2FEM1B-DEL | 47.57 ± 0.25 |
| CRL2FEM1B-F549R | 50.51 ± 0.03 |
| CRL2FEM1B-F549S | 50.4 ± 0.13 |
| CRL2FEM1B-F549T | 53.78 ± 0.03 |
| CRL2FEM1B-Y275R/L278R | 48.91 ± 0.05 |
| MBP-FCB-WT | 46.78 ± 0.01 |
| MBP-FCB-DEL | 47.23 ± 0.05 |
| MBP-FCB-F549R | 46.58 ± 0.02 |
| MBP-FCB-F549S | 46.69 ± 0.02 |
| MBP-FCB-F549T | 46.7 ± 0.02 |
| MBP-FCB-Y275R/L278R | 47.69 ± 0.02 |

**Table 2.** Molecular weights of wild-type and mutants of (N8-) CRL2FEM1B and E3-substrate complexes measured by mass photometry experiments.

| Sample | Molecular weight (kDa) | | |
|---|---|---|---|
| CRL2FEM1B-WT | 191 ± 1 | 414 ± 31 | 576 ± 3 |
| CRL2FEM1B-DEL | 191 ± 9 | | |
| CRL2FEM1B-F549R | 189 ± 3 | 386 ± 13 | |
| CRL2FEM1B-F549T | 186 ± 0 | 384 ± 21 | |
| CRL2FEM1B-F549S | 187 ± 3 | 375 ± 11 | |
| CRL2FEM1B-Y275RL278R | 188 ± 11 | | |
| N8-CRL2FEM1B-WT | 197 ± 5 | 402 ± 13 | |
| N8-CRL2FEM1B-DEL | 197 ± 27 | | |
| N8-CRL2FEM1B-F549R | 203 ± 28 | | |
| N8-CRL2FEM1B-F549T | 190 ± 18 | 395 ± 11 | |
| N8-CRL2FEM1B-F549S | 192 ± 17 | | |
| N8-CRL2FEM1B-Y275RL278R | 183 ± 7 | | |
| N8-CRL2FEM1B-WT-BEX2 | | 517 ± 36 | |
| N8-CRL2FEM1B-WT-FNIP1/FLCN | | 380 ± 12 | |

wild-type CRL2$^{FEM1B}$ existed as trimer, dimer, and monomer in solution (Table 2; Appendix Fig. S8A). Oligomerization-deficient mutant FEM1B$^{DEL}$ and FEM1B$^{Y275R/L278R}$ existed as monomer while FEM1B$^{F549T}$, FEM1B$^{F549S}$, and FEM1B$^{F549R}$ existed as monomer and dimer (Table 2; Appendix Fig. S8B–F). These results further confirmed the oligomer interfaces shown in our structure (Appendix Fig. S8A–F).

## Neddylation alters the oligomeric state of CRL2$^{FEM1B}$ E3 ubiquitin ligase

Our oligomeric CRL2$^{FEM1B}$ super-complex shows that the WHB domain plays an important role in its assembly. Because the ubiquitin-like protein NEDD8 is conjugated to Lys689 in the flexible WHB domain (residues 661–745) of the CUL2 scaffold protein (Wada et al, 1999), We hypothesized that CRL2$^{FEM1B}$ neddylation might impair the protomer 3-protomer 2 interaction, as protomer 3 bound protomer 2 weakly in a 'head-to-tail' orientation through the WHB domain and RBX1 (Fig. 4A). To further investigate the oligomeric state of neddylated CRL2$^{FEM1B}$ (N8-CRL2$^{FEM1B}$), we reconstituted the fully assembled complex in vitro and then neddylation reactions were performed to generate the N8-CRL2$^{FEM1B}$ complex (Fig. 4B). By using blue native PAGE together with glycerol gradient ultracentrifugation and MP experiment, we obtained evidence indicating that the trimeric CRL2$^{FEM1B}$ disassociates into a dimeric conformation after neddylation (Fig. 4C,D; Appendix Fig. S8G). Taken together, our results show that the oligomeric state of CRL2$^{FEM1B}$ is altered by neddylation.

## BEX2 and FNIP1/FLCN are ubiquitylated by N8-CRL2$^{FEM1B}$ in vitro

To study the enzyme activity of N8-CRL2$^{FEM1B}$, we purified its substrates MBP-FNIP1$^{degron}$, FNIP1/FLCN, and MBP-BEX2 (Manford et al, 2021; Manford et al, 2020) and performed ubiquitylation assays in vitro. N8-CRL2$^{FEM1B}$ had E3 ubiquitin ligase activities towards MBP-BEX2, MBP-FNIP1$^{degron}$, and FNIP1/FLCN according to our experiment results (Fig. 5A–C). In view of previous reports

about the functions of CRL oligomerization and the oligomeric state of CRL2$^{FEM1B}$, we compared molecular weights of neddylated CRL2$^{FEM1B}$, MBP-BEX2-N8-CRL2$^{FEM1B}$ and MBP-FNIP1$^{degron}$-N8-CRL2$^{FEM1B}$ by blue native PAGE, and measured the molecular weight of MBP-BEX2-N8-CRL2$^{FEM1B}$ by MP experiments. Then, we performed SEC-MALS analysis and MP experiment on N8-CRL2$^{FEM1B}$-FNIP1-FLCN. Blue native PAGE showed that MBP-FNIP1$^{degron}$-N8-CRL2$^{FEM1B}$ and MBP-BEX2-N8-CRL2$^{FEM1B}$ had a substantially greater molecular weight than N8-CRL2$^{FEM1B}$ dimer (Fig. 5D). According to SEC-MALS analysis and MP experiment, N8-CRL2$^{FEM1B}$-FNIP1-FLCN has a molecular weight of 427.7 kDa and 380 kDa, respectively, which corresponds to a combined calculated molecular weight of monomeric N8-CRL2$^{FEM1B}$ and monomeric FNIP1/FLCN heterodimer (201.4 kDa and 195.8 kDa, respectively) (Table 2; Fig. 5E; Appendix Fig. S8N). MBP-BEX2-N8-CRL2$^{FEM1B}$ has a molecular weight of 517 kDa, which corresponds to a combined calculated molecular weight of dimeric N8-CRL2$^{FEM1B}$ with two MBP-BEX2 molecules (402.8 kDa and 58.5 kDa, respectively) (Table 2; Appendix Fig. S8M). These results suggest that BEX2 and FNIP1$^{degron}$ are ubiquitylated by dimeric N8-CRL2$^{FEM1B}$ while FNIP1/FLCN is ubiquitylated by monomeric N8-CRL2$^{FEM1B}$.

## Cryo-EM structure of N8-CRL2$^{FEM1B}$-BEX2 complex

To investigate the structural basis of ubiquitylation of BEX2 by neddylated CRL2$^{FEM1B}$, we determined the cryo-EM structure of N8-CRL2$^{FEM1B}$ complexed with MBP-BEX2 (Appendix Table S1, Appendix Figs. S3A,B and S5A). Unexpectedly, two protomers of N8-CRL2$^{FEM1B}$ in the N8-CRL2$^{FEM1B}$-BEX2 complex adopted similar conformation comparing to protomer 1 and protomer 2 of unneddylated CRL2$^{FEM1B}$ (Fig. 6A). We docked the structure of unneddylated CRL2$^{FEM1B}$ with a high degree of confidence and adjusted the model according to the densities. We aligned protomer 1 and protomer 2 of unneddylated CRL2$^{FEM1B}$ with the model of N8-CRL2$^{FEM1B}$-BEX2 with Cα RMSD values of 2.152 (2170 to 2170

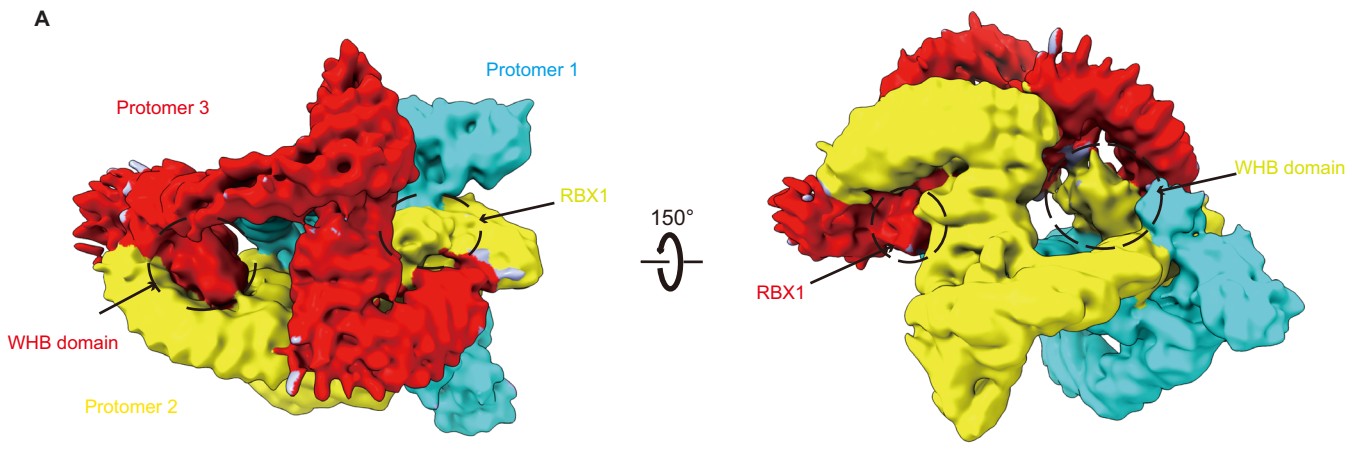

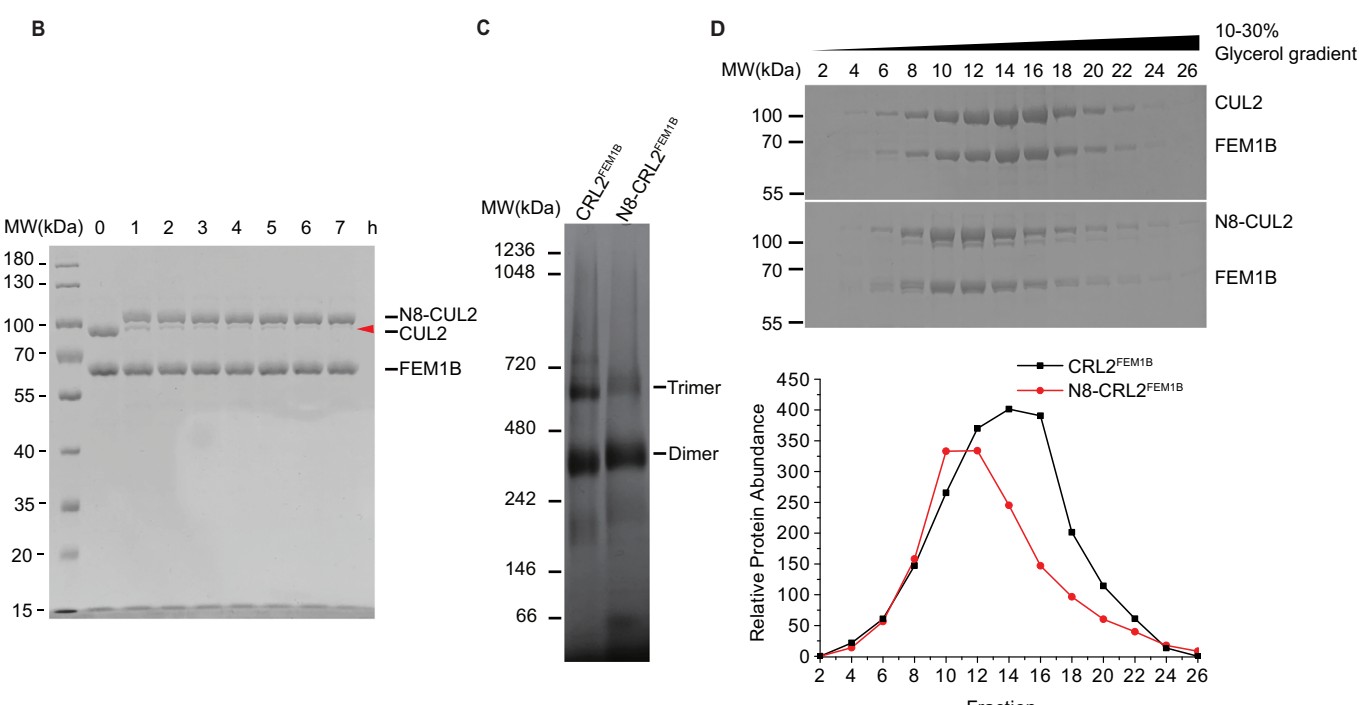

**Figure 4. Neddylation regulates the oligomerization states of CRL2^FEM1B E3 ubiquitin ligase.**

(A) Protomers 2 and 3 form contacts in a 'head' to 'tail' conformation in which the substrate recognition pocket of one protomer is occupied by the RBX1 and flexible WHB domain of CUL2 of the other protomer. Broken black circled regions correspond to RBX1 subunit and WHB domain of protomer 2 and protomer 3, respectively. (B) In vitro neddylation of CRL2^FEM1B shows CUL2 are completely neddylated after 4 h as the band of unneddylated CUL2 (marked by red triangle) are nearly invisible. Samples were resolved by SDS-PAGE and stained with Coomassie brilliant blue. (C) Neddylation induces disassociation of trimeric CRL2^FEM1B and formation of dimeric N8-CRL2^FEM1B. Samples were analyzed by blue native PAGE and stained with Coomassie brilliant blue. (D) Fractions of CRL2^FEM1B with unneddylated and neddylated CUL2 were separated by glycerol gradient ultracentrifugation. Samples were resolved by SDS-PAGE and stained with Coomassie brilliant blue. Source data are available online for this figure.

atoms) (Fig. 6B). However, interacting with BEX2 induced an outward movement of the N-terminal domain of FEM1B of protomer 1 comparing to protomer 1 of unneddylated CRL2^FEM1B (Fig. 6C,D).

The 'forceps'-like conformation formed by C-terminal WHB domain of CUL2 and RBX1 in protomer 1 interacted with the cullin repeat 3 of CUL2 in protomer 2, while the N-terminal domain α-

helix of FEM1B (residues 269–284) in protomer 1 interacted with the C-terminal domain ankyrin repeats (residues 546–588) of FEM1B in protomer 2 (Fig. 6E,F). Compared to protomer 2, protomer 1 was relatively stable acting as the scaffold. According to the results of 3D variability analysis, the catalytic domain of protomer 2 was moving up and down around the substrate recognition domain of protomer 1, allowing it to adjust the distance

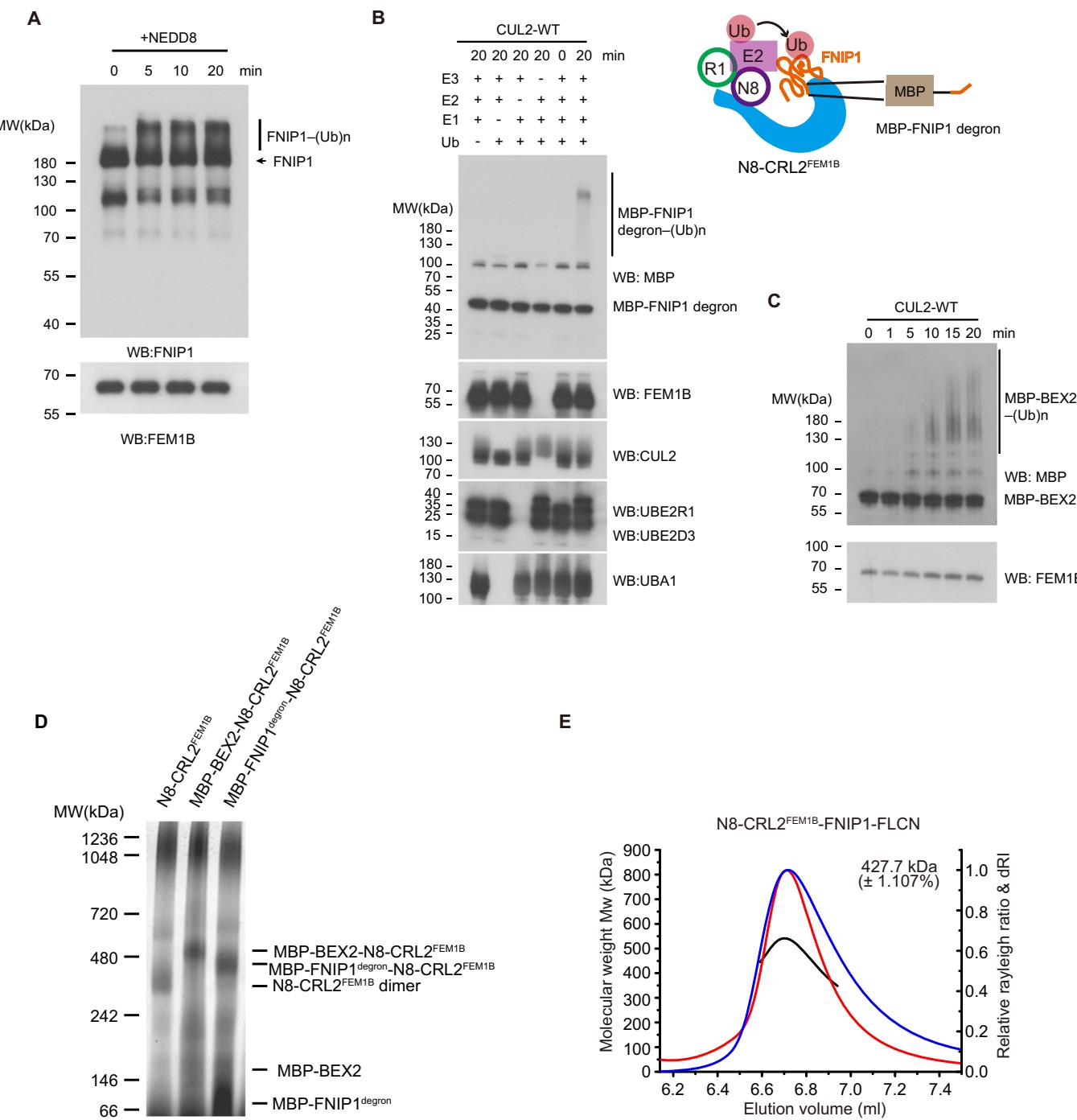

**Figure 5. BEX2 interacts with dimeric N8-CRL2^FEM1B while FNIP1/FLCN binds to monomeric N8-CRL2^FEM1B.**

(A) In vitro ubiquitylation of FNIP1/FLCN by N8-CRL2^FEM1B. Samples were analyzed by western blotting using antibodies against FNIP1 and FEM1B. (B) Drop out experiments of in vitro ubiquitylation of MBP-FNIP1^degron. Samples were ubiquitylated in vitro and then analyzed by western blotting using antibodies against MBP, FEM1B, CUL2, UBE2R1, UBE2D3, and UBA1. The construct is shown in cartoon representations. CUL2-ELOB-ELOC-FEM1B, RBX1, NEDD8, Ub, E2, and FNIP1 are colored in cyan, green, purple, pink, magenta, and orange, respectively. (C) In vitro ubiquitylation of MBP-BEX2 by N8-CRL2^FEM1B. Samples were analyzed by western blotting using antibodies against MBP and FEM1B. (D) MBP-BEX2 and MBP-FNIP1^degron bind to dimeric N8-CRL2^FEM1B. Bands are indicated by dash lines, respectively. Samples were analyzed by blue native PAGE and stained with Coomassie brilliant blue. (E) SEC-MALS analysis shows that neddylated CRL2^FEM1B (MW 192.8 kDa) binds FNIP1/FLCN (MW 195.8 kDa) as a monomer at a ratio of 1:1. The red, blue, and black lines indicate the relative Rayleigh ratio, relative dRI ratio, and molecular weight, respectively. Source data are available online for this figure.

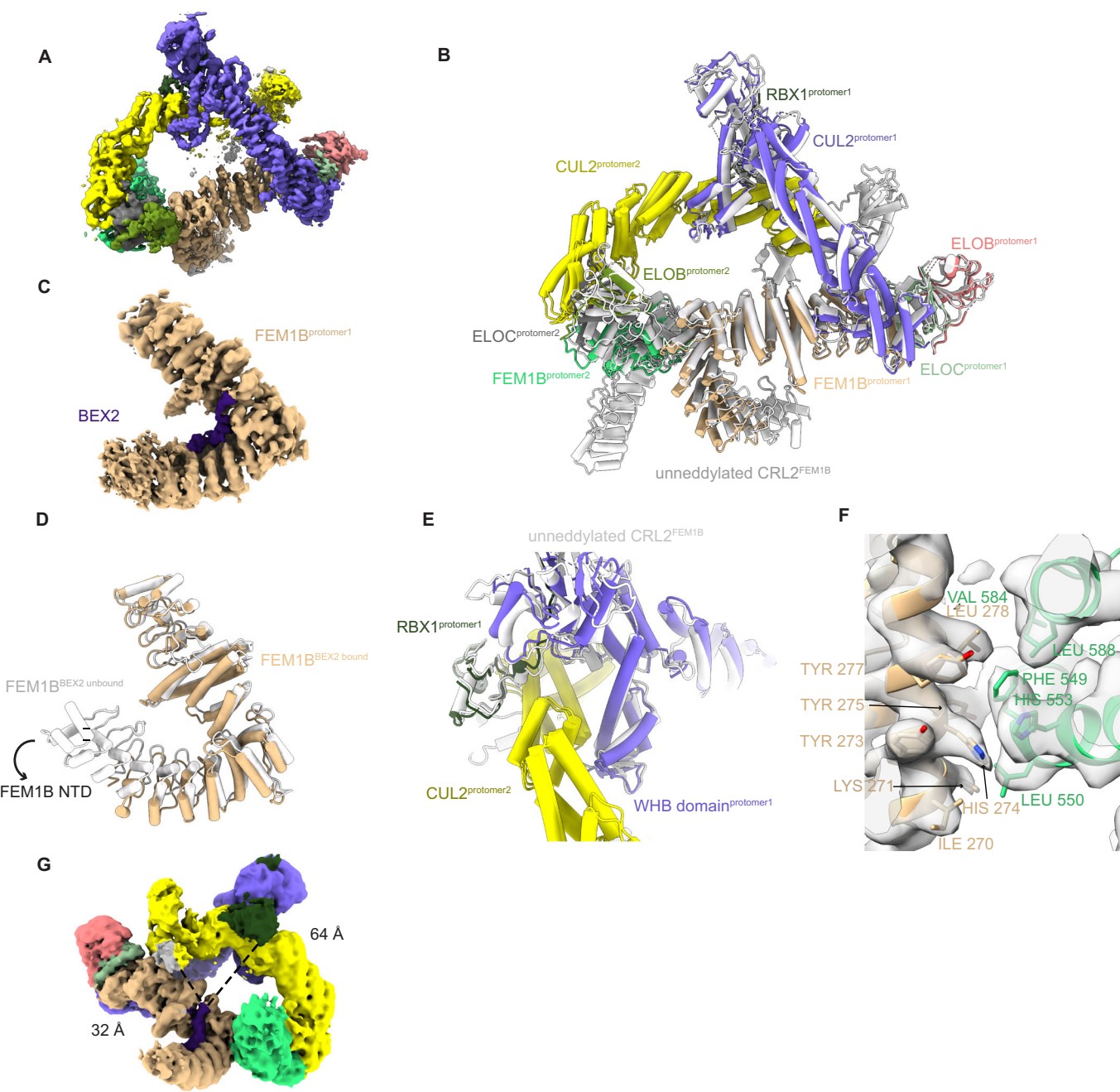

**Figure 6. N8-CRL2^FEM1B^-BEX2 complex adopts a similar conformation as unneddylated dimeric CRL2^FEM1B^ complex.**

(A) Cryo-EM densities of N8-CRL2^FEM1B^-BEX2 complex with each of its subunits assigned a different color as shown in (B). (B) Atomic model of N8-CRL2^FEM1B^-BEX2 complex with each of its subunits assigned a different color superimposed on atomic model of unneddylated CRL2^FEM1B^ complex (colored in white). (C) Cryo-EM densities of FEM1B in protomer 1 colored according to (A) and BEX2 (purple). (D) BEX2 induces an outward movement of the N-terminal domain of FEM1B of protomer 1 shown by aligning holo-FEM1B to apo-FEM1B. (E) Similar to unneddylated CRL2^FEM1B^ complex, the cullin repeat of CUL2 in protomer 2 is clamped by a 'forceps' conformation formed by RBX1 and the WHB domain of CUL2 in protomer 1 in N8-CRL2^FEM1B^-BEX2 complex. (F) Atomic models fitted in cryo-EM map shows details of the hydrophobic interactions between the N-terminal α-helix of FEM1B (residues 269–284) in protomer 1 and the C-terminal ankyrin repeats of FEM1B (residues 546–588) in protomer 2. (G) Dimeric conformation shortens about half of the distances between substrates and catalytic domain comparing to monomeric conformation.

between E2~Ub and substrates (Appendix Fig. S10B). Under this circumstance, we believe that FEM1B of protomer 1 serves as the only substrate recognition domain in this complex while RBX1 and NEDD8 of protomer 2 act as the only catalytic domain, because RBX1 of protomer 1 and FEM1B of protomer 2 are occluded. This asymmetric dimeric conformation provides a much closer catalytic domain to substrates, comparing to monomeric CRL2^FEM1B^ (Fig. 6G).

To investigate if dimerization of N8-CRL2^FEM1B^ is critical for ubiquitylation of BEX2, we again performed experiments on the

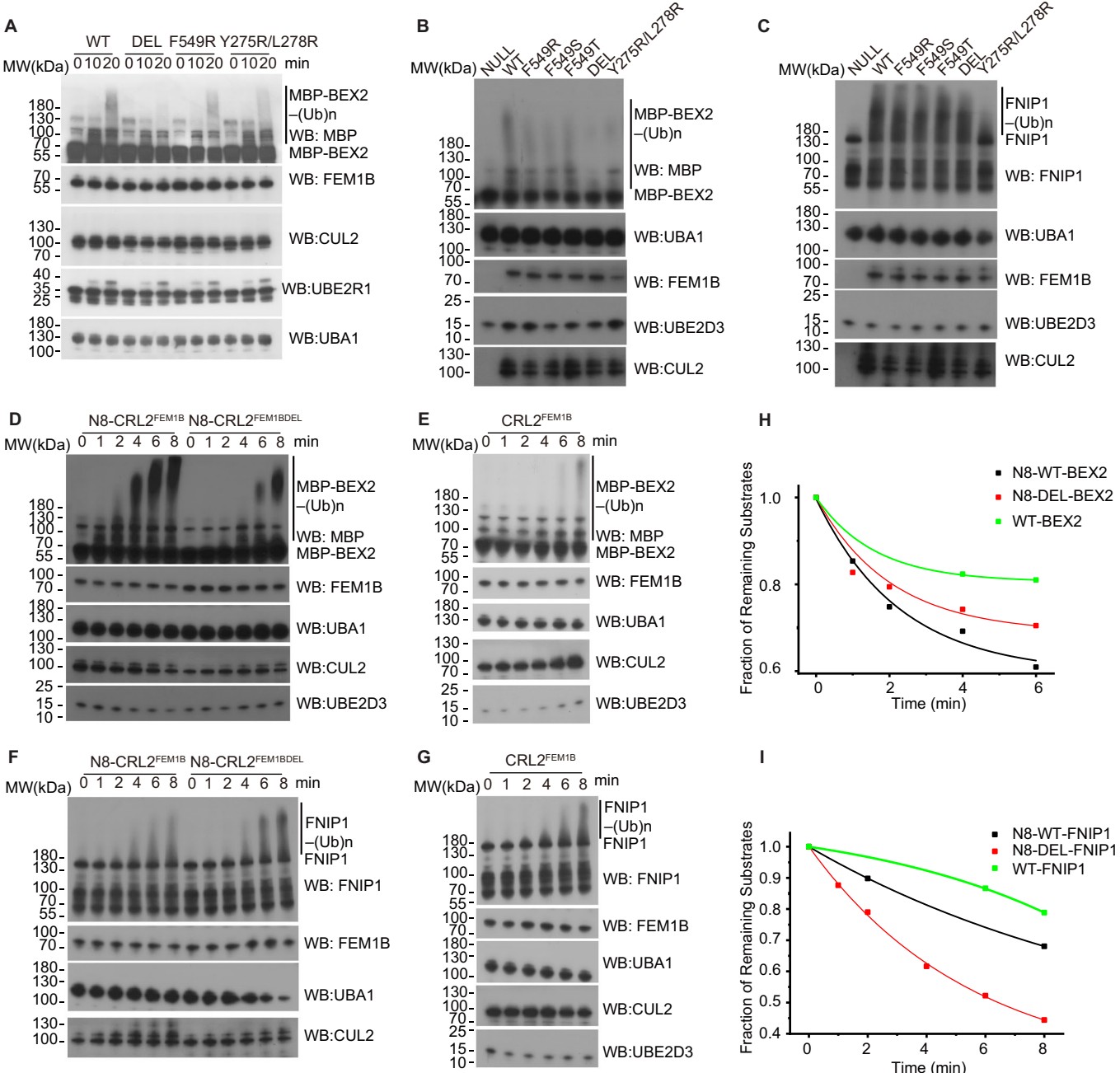

**Figure 7. BEX2 is ubiquitylated by dimeric N8-CRL2^FEM1B while FNIP1/FLCN is ubiquitylated by monomeric N8-CRL2^FEM1B.**

(A,B) The results of in vitro ubiquitylation of BEX2 by wild-type N8-CRL2^FEM1B and its oligomerization-deficient mutants suggest dimerization of N8-CRL2^FEM1B is necessary for ubiquitylation of BEX2. Samples were analyzed by western blotting using antibodies against MBP, FEM1B, CUL2, UBA1, UBE2D3, or UBE2R1. (C) The results of in vitro ubiquitylation of FNIP1/FLCN by wild-type N8-CRL2^FEM1B and its oligomerization-deficient mutants suggest mutants F549R, F549T, F549S, and DEL have no negative impact on the enzyme activity while Y275R/L278R shows lower ubiquitylation efficiency towards FNIP1/FLCN. (D–G) Time-resolved in vitro ubiquitylation assays of MBP-BEX2 and FNIP1 by N8-CRL2^FEM1B, N8-CRL2^FEM1BDEL, and CRL2^FEM1B show that N8-CRL2^FEM1B has higher enzyme activity towards MBP-BEX2 than N8-CRL2^FEM1BDEL, while N8-CRL2^FEM1BDEL is more active to FNIP1 comparing to N8-CRL2^FEM1B. However, CRL2^FEM1B has minor enzyme activity towards either MBP-BEX2 or FNIP1. (H,I) Plots of the fraction of substrates that have not been converted to ubiquitylated products against time for ubiquitylation reactions containing CRL2^FEM1B, N8-CRL2^FEM1B, or N8-CRL2^FEM1BDEL (as shown in (D–G)). The plots of ubiquitylated substrates were quantified using ImageJ. The data were fit to one-phase exponential decay function with time constant parameter ($y = A1*exp(-x/t1)+y0$) to obtain rates of ubiquitylation. Source data are available online for this figure.

**Table 3. The effect of mutations of CRL2FEM1B on binding affinities of substrates.**

| Substrate | Sample | Binding affinity |
|---|---|---|
| FNIP1 | CRL2FEM1B-WT | 46.254 nM |
| | CRL2FEM1B-DEL | 4.719 nM |
| | CRL2FEM1B-F549R | 8.987 nM |
| | CRL2FEM1B-F549S | 17.485 nM |
| | CRL2FEM1B-F549T | 33.279 nM |
| | CRL2FEM1B-Y275R/L278R | 3.126 nM |
| BEX2 | CRL2FEM1B-WT | 44.666 nM |
| | CRL2FEM1B-DEL | 6.624 nM |
| | CRL2FEM1B-F549R | 14.024 nM |
| | CRL2FEM1B-F549S | 22.963 nM |
| | CRL2FEM1B-F549T | 109.396 nM |
| | CRL2FEM1B-Y275R/L278R | 3.990 nM |

oligomerization-deficient mutants $FEM1B^{F549R}$, $FEM1B^{F549S}$, $FEM1B^{F549T}$, $FEM1B^{Y275R/L278R}$, and $FEM1B^{\Delta546-553}$ ($FEM1B^{DEL}$), as $N8\text{-}CRL2^{FEM1B}\text{-}BEX2$ complex adopted the similar conformation as protomer 1 and protomer 2 of $CRL2^{FEM1B}$. Consistent with our expectation, MP experiment results indicated that $CRL2^{FEM1B}$ oligomerization-deficient mutants mainly formed monomers in the solution except $FEM1B^{F549T}$ (Table 2; Appendix Fig. S8H–L). We then performed in vitro ubiquitylation assays at room temperature for 20 min and assessed the ubiquitylation level of MBP-BEX2 and FNIP1/FLCN with $N8\text{-}CRL2^{FEM1B}$ and neddylated mutants by western blotting using MBP and FNIP1 antibodies. The in vitro ubiquitylation experiments revealed that oligomerization-deficient mutants except $FEM1B^{F549T}$ had significantly lower ubiquitylation activities toward BEX2 than wild-type $N8\text{-}CRL2^{FEM1B}$, whereas these mutants did not affect the ubiquitylation of FNIP1/FLCN significantly (Fig. 7A–C). We also assessed the time-resolved ubiquitylation level of MBP-BEX2 and FNIP1/FLCN with $N8\text{-}CRL2^{FEM1B}$, $N8\text{-}CRL2^{FEM1BDEL}$, and $CRL2^{FEM1B}$, respectively (Fig. 7D–G). The plots from the chromogenic western blotting analysis of the ubiquitylated products of MBP-BEX2 confirmed that $N8\text{-}CRL2^{FEM1B}$ had higher enzyme activity than $N8\text{-}CRL2^{FEM1BDEL}$ and $CRL2^{FEM1B}$ toward MBP-BEX2 (Fig. 7H). As expected, the plots from the chromogenic western blotting analysis of the ubiquitylated products of FNIP1/FLCN indicated that the oligomerization-deficient mutant $N8\text{-}CRL2^{FEM1BDEL}$ had higher enzyme activity than $N8\text{-}CRL2^{FEM1B}$ and $CRL2^{FEM1B}$ (Fig. 7I). Then we performed grating-coupled interferometry experiments on oligomerization-deficient mutants and wild-type $CRL2^{FEM1B}$. The results suggested that mutations didn't influence the process of substrate recognition (Table 3; Appendix Fig. S9). Taken together, the interaction with dimeric $N8\text{-}CRL2^{FEM1B}$ is required for the ubiquitylation of BEX2.

### Cryo-EM structure of N8-CRL2$^{FEM1B}$-FNIP1-FLCN complex

To determine whether the ubiquitylation strategy of this asymmetric $N8\text{-}CRL2^{FEM1B}$ dimer is applicable to other substrates, we obtained the stable $N8\text{-}CRL2^{FEM1B}\text{-}FNIP1\text{-}FLCN$ complex and its cryo-EM structure (Appendix Table S1, Appendix Figs. S4A,B and S5A). We docked protomer 1 of unneddylated $CRL2^{FEM1B}$ and the recently published crystal structure of $FEM1B\text{-}FNIP1^{degron}$ (PDB:7ROY) (Manford et al, 2021) into the map (Fig. 8A). We compared our structure with a previously reported structure of unneddylated $CRL2^{FEM1B}$ and noted that neddylated $CRL2^{FEM1B}$-FNIP1-FLCN exhibits an additional density in the canonical FEM1B substrate binding site. This observation is consistent with the recently reported structure of FEM1B complexed with $FNIP1^{degron}$ (Manford et al, 2021; Manford et al, 2020) (Fig. 8B). The map also shows that the FNIP1/FLCN heterodimer forms multiple contacts with CUL2-RBX1 and FEM1B. While the neddylated WHB domain of CUL2 and RBX1 provides a platform for the E2s and fixes one side of the substrate, FEM1B (as a substrate receptor) has three additional major binding sites for the other side of the substrate. In addition to the central part of $FNIP1^{degron}$ (residues 579–585) forming $Zn^{2+}$-binding motifs with the FEM1B N-terminus as recently reported (Manford et al, 2021), FNIP1 (residues 568–574) also interacts with the α-helical region in the FEM1B C-terminus (Fig. 8C). Furthermore, the N-terminus (residues 1–26) and the C-terminus (residues 536–593) of FEM1B appear to be involved in binding the FNIP1/FLCN complex (Fig. 8D,E). These results support the following hypothesis: multiple regions of interactions between $N8\text{-}CRL2^{FEM1B}$ and FNIP1/FLCN complex are needed to orient the FNIP1/FLCN complex relative to $N8\text{-}CRL2^{FEM1B}$, and fill in the gap between RBX1 and FEM1B. Thus, FNIP1/FLCN is efficiently ubiquitinated by monomeric $N8\text{-}CRL2^{FEM1B}$.

## Discussion

In the present study, we have demonstrated the ubiquitylation strategy of oligomeric $CRL2^{FEM1B}$ E3 ubiquitin ligase according to substrates with different scales (Fig. 8F). For proteins with low molecular weight like BEX2, $N8\text{-}CRL2^{FEM1B}$ adopts an asymmetric dimeric conformation. In this state, one protomer serves as scaffold and provides substrate recognition domain, it remains inactive because the catalytic subunit RBX1 is occluded. The other protomer provides the catalytic subunit but its substrate recognition subunit FEM1B is occluded. Thus, this conformation shows a novel approach to shorten the gap between catalytic subunit and substrate recognition subunit of CRLs. For proteins with large scales like FNIP1/FLCN complex, substrates capture the monomeric state of $N8\text{-}CRL2^{FEM1B}$ and fill in the gap between RBX1 and FEM1B. To the best of our knowledge, our data represent the first example of the asymmetric homo-dimerization of CRL E3 ubiquitin ligase and shed the lights on the importance of $CRL2^{FEM1B}$ oligomerization.

A large body of evidence suggests that many CRLs oligomerize via a variety of mechanisms, the majority of which are mediated by their substrate-recognition subunits. For example, the F-box containing protein FBW7 (the substrate recognition subunit of the E3 ubiquitin ligase $CRL1^{FBW7}$), forms oligomers via its N-terminal D domain, and oligomerization enhances ubiquitination and processivity (Hao et al, 2007). However, how the D domain of FBW7 mediates oligomerization is not known. Mutations of several hydrophobic residues in the D domain abolish its oligomerization (Hao et al, 2007), indicating that it may oligomerize via hydrophobic interactions as FEM1B does in our structure of its unneddylated state. In another example of CRL oligomerization, KEAP1, a substrate-specific adapter of a BCR

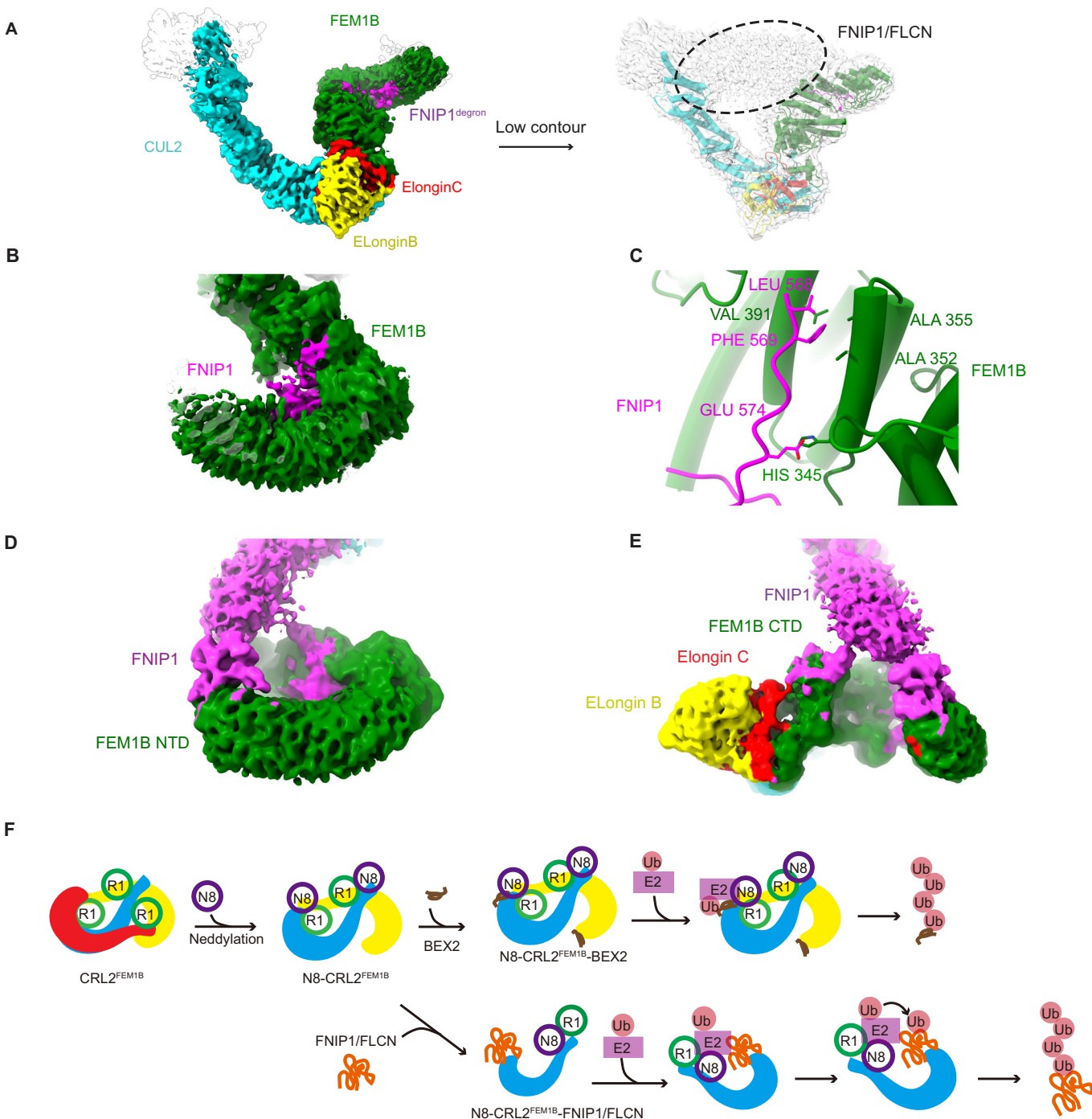

**Figure 8.  Multiple regions of interactions between N8-CRL2^FEM1B and FNIP1/FLCN complex for substrate recognition.**

(**A**) Cryo-EM map of the N8-CRL2^FEM1B-FNIP1/FLCN complex at high and low contour. Each of the five components of CRL2^FEM1B along with FNIP1^degron are assigned a different color the same as Fig. 1E. The density for the FNIP1/FLCN complex is circled in the low contour map. (**B**) FNIP1^degron (PDB: 7ROY, magenta) fits well in an extra density found inside the substrate binding pocket of FEM1B (green) in the FNIP1/FLCN bound state as shown in the densities of the N8-CRL2^FEM1B-FNIP1/FLCN complex. (**C**) Atomic details of the interaction between FNIP1^degron (568–574) colored in magenta and the α-helical region (345–391) of FEM1B (green). (**D**) Cryo-EM density representing the interaction between the FNIP1/FLCN complex (magenta) and the N-terminal domain of FEM1B (green). (**E**) Cryo-EM density representing the interaction between the FNIP1/FLCN complex (magenta) and the C-terminal domain of FEM1B (green). (**F**) Model of the activation and substrate recognition process of CRL2^FEM1B. Each promoter is colored separately in cyan, yellow, and red. RBX1, NEDD8, Ub, E2, and substrates are colored in green, purple, pink, magenta, brown, and orange, respectively.

(BTB-CUL3-RBX1) E3 ubiquitin ligase complex, dimerizes via its BTB domain which facilitates cullin-mediated ubiquitylation of proteins by a 'tethering' mechanism where two sites of the substrate Nrf2 subsequently bind two KEAP1 copies in the KEAP1 homodimer (McMahon et al, 2006). Extensive conservation of the BTB dimer interface of approximately 200 human BTB proteins, however, leads to dysfunctional heterodimerization, which is regulated by the dimerization quality control E3 ligase, SCF-FBXL17 (Mena et al, 2020). Thus SCF–FBXL17 ubiquitylates and aids the degradation of inactive BTB protein heterodimers while leaving functional homodimers intact (Mena et al, 2020). In this way, KEAP1 homo-dimerization is essential for its normal activity as well as its stability, and aberrant BTB dimers are degraded by SCF-FBXL17.

In addition to the substrate receptor, other components of CRLs contribute to their oligomerization. Structural and biochemical studies of CRL4[DCAF1] (the CUL4-RBX1-DDB1-DCAF1 complex) revealed that unneddylated substrate-free CRL4[DCAF1] is an inactive tetramer mediated by the DCAF1 WD40 and LisH homology domains, as well as the C-terminal domain (CTD) of CUL4 and RBX1 (Mohamed et al, 2021). Neddylation of CRL4[DCAF1] disrupts the interaction between the CUL4 CTD and the neighboring DCAF1 protomer, resulting in an active dimeric conformation that is favored by its viral substrates (Mohamed et al, 2021). Our findings serendipitously reveal an oligomerization mechanism in which CRL2[FEM1B] controls its activity via oligomerization mediated by both its substrate acceptor FEM1B and CUL2-RBX1. Thus, the oligomerization mechanisms of CRLs are diverse and can play an important role in adapting to substrates of different sizes. However, more experiments are needed to explain the structure basis of the disassociation of dimeric N8-CRL2[FEM1B] induced by FNIP1/FLCN complex.

Taken together, our findings shed light on the structural basis of mechanism of CRLs which has important implications for fully understanding the roles of CRLs in many different signaling pathways and biological processes. In addition, because FEM1B has been identified as a promising candidate for proteolysis-targeting chimeras (PROTAC) (Henning et al, 2022), a better understanding of CRL2[FEM1B] may prove beneficial to its development as a therapeutic target.

# Methods

## Reagents and tools

See Table 4.

**Table 4. Reagents and tools.**

| REAGENT or RESOURCE | SOURCE | IDENTIFIER |
|---|---|---|
| Antibodies | | |
| Rabbit monoclonal anti-MBP | Abcam | Cat#ab119994; RRID: AB_10900967 |
| Rabbit monoclonal anti-FNIP1 | Abcam | Cat#ab134969; EPNCIR107 |
| Rabbit polyclonal anti-FEM1B | Abclonal | Cat#A12802; RRID: AB_2759641 |
| Bacterial and Virus Strains | | |
| BL21(DE3) Chemically Competent Cell | TransGen Biotech | CD701-02 |
| Trans5α Chemically Competent Cell | TransGen Biotech | CD201-01 |
| Chemicals, Peptides, and Recombinant Proteins | | |
| Tris(2-carboxyethyl)phosphine (TCEP) | Hampton | HR2-801 |
| Dithiothreitol (DTT) | INALCO | 1758-9030-25g |
| Isopropyl β-D-Thiogalactoside (IPTG) | INALCO | 1758-1400-25g |
| Phenylmethylsulfonyl fluoride (PMSF) | Sigma-Aldrich | P7626-25G |
| Imidazole | Sigma-Aldrich | V90015-500g |
| HEPES | Sigma-Aldrich | H3375-1kg |
| Magnesium Chloride | Sigma-Aldrich | M2670-500g |
| Sodium Chloride | Sigma-Aldrich | V90058 |
| Tris | Lablead | V900483 |
| Recombinant MBP-FEM1B/ELOB/ELOC | This paper | N/A |
| Recombinant cHIS-CUL2/RBX1 | This paper | N/A |
| Recombinant UBA3/NAE1 | This paper | N/A |
| Recombinant UBE2M | This paper | N/A |
| Recombinant UBE2D3 | This paper | N/A |
| Recombinant UBE2R1 | This paper | N/A |
| Recombinant Ubiquitin | This paper | N/A |
| Recombinant NEDD8 | This paper | N/A |

**Table 4.** (continued)

| REAGENT or RESOURCE | SOURCE | IDENTIFIER |
|---|---|---|
| Recombinant UBA1 | This paper | N/A |
| Recombinant FNIP1-FLCN | This paper | N/A |
| Experimental Models: Cell Lines | | |
| SF9 insect cell | Invitrogen | |
| FreeStyle 293-F Cells | ThermoFischer Scientific | Cat. # R79007 |
| HEK293T | ThermoFischer Scientific | |
| Recombinant DNA | | |
| pET-28a(+)-6*HIS-MBP-TEV-FEM1B | This paper | N/A |
| pETDuet 1-ELOB-ELOC(17-112) | This paper | N/A |
| pETDuet 1-NAE1 | This paper | N/A |
| pET-28a(+)-6*HIS-GST-TEV-UBA3 | This paper | N/A |
| pET-28a(+)-6*HIS-TEV-UBE2M | This paper | N/A |
| pET-28a(+)-6*HIS-TEV-UBE2D3 | This paper | N/A |
| pET-28a(+)-6*HIS-TEV-UBE2R1 | This paper | N/A |
| pET-28a(+)-6*HIS-TEV-Ubiquitin | This paper | N/A |
| pET-28a(+)-6*HIS-TEV-NEDD8(1–76) | This paper | N/A |
| pFastBac HTB-6*HIS-UBA1 | This paper | N/A |
| pFastBac Dual-cHIS-CUL2-RBX1 | This paper | N/A |
| pFastBac Dual-cHIS-CUL2-N355R/ C465R-RBX1 | This paper | N/A |
| pFastBac Dual-CMV-FNIP1-mCherry-6*HIS-StrepII -EF1α-FLCN | This paper | N/A |
| pCDH-S-tag-HA-FEM1B | This paper | N/A |
| pCDH-FLAG-FEM1B | This paper | N/A |
| pET-28a(+)-6*HIS-MBP-TEV-BEX2 | This paper | N/A |
| pET-28a(+)-6*HIS-MBP-TEV-FNIP1degron | This paper | N/A |
| Primers | | |
| CUL2-f | This paper | CCCACCATCGGGCGCGCCACCatgtctttgaaaccaagagtagtag |
| CUL2-r | This paper | GTGATGGTGATGATGCgcgcgacgtagctgtattcatc |
| RBX1-f | This paper | TGATCACCCGGGATCTCGAGatggcggcagcgatggatg |
| RBX1-r | This paper | ATCAGCTGCTAGCACCATGGctagtgcccatacttttgg |
| FEM1B-f | This paper | GAAAACCTGTATTTTCAGGGCGAATTCATGGAGGGCCTGGCTGGCTAT |
| FEM1B-r | This paper | GGTGGTGGTGCTCGAGTTAATGAAATCCAACAAACTCTTCAAGAGTTC |
| FLCN-2nd-f | This paper | CTAGCGCTACCGGTCGCCACCATGAATGCCATCGT |
| FLCN-2nd-r | This paper | CCATCTCCCGGTACCTCAGTTCCGAGACTCCGAGGC |
| FNIP1-2nd-f | This paper | CTCTTAAGGGAATTCGCCACCatggcccctacg |
| FNIP1-2nd-r | This paper | GAAAATACAGGTTTTCaaggagtatttgtgcaacatatggagagt |
| degron-f | This paper | CTGTATTTTCAGGGCAACAAATCTTCTCTGCTGTTCAAAGAATCTGAAG |
| degron-r | This paper | CAGTGGTGGTGGTGGTGGTGCTCGATCACTGACC |
| NEDD8-BamHI-f | This paper | GCAAATGGGTCGCGGATCCATGCTAATTAAAGTGAAGACGC |
| NEDD8-NotI-r | This paper | CTCGAGTGCGGCCGCTCACTGCCTAAGACCACC |
| GST-UBA3-f | This paper | GTATTTTCAGGGAGAATTCATGGCGGATGGCGAGGAGCC |
| GST-UBA3-r | This paper | GTGGTGGTGCTCGAGTTAAGAAGTAAAATGAAGTTTGAATAGTACAGTCTGTGGGG |
| NAE1-f | This paper | GGAGATATACATATGgcgcagctgggaaagctgctc |

**Table 4.** (continued)

| REAGENT or RESOURCE | SOURCE | IDENTIFIER |
|---|---|---|
| NAE1-r | This paper | CTTTACCAGACTCGAGctacaactggaaagttgctgaagtttgtgacatg |
| UBE2M-BamHI-f | This paper | GCAAATGGGTCGCGGATCCATGATCAAGCTGTTCTCG |
| UBE2M-NotI-r | This paper | CTCGAGTGCGGCCGCCTATTTCAGGCAGCGC |
| UBA1-BamH1-f | This paper | GCAAATGGGTCGCGGATCCATGTCCAGCTCGCCGCTG |
| UBA1-NotI-r | This paper | CTCGAGTGCGGCCGCTCAGCGGATGGTGTATCGGAC |
| UBE2D3-BamHI-f | This paper | CAGCAAATGGGTCGCATGGCGCTGAAACGGATTAATAAG |
| UBE2D3-NotI-r | This paper | GTGCTCGAGTGCGGCCTCACATGGCATACTTCTGAGTCC |
| Y275L278/S-F | This paper | CATAAAGACATACCACTCTCTATATTCAGCCATGTTAGAGAGGTTCC |
| Y275L278/S-R | This paper | GGAACCTCTCTAACATGGCTGAATATAGAGAGTGGTATGTCTTTATG |
| Y275L278/R-F | This paper | CATAAAGACATACCACCGTCTATATCGAGCCATGTTAGAGAGGT |
| Y275L278/R-R | This paper | ACCTCTCTAACATGGCTCGATATAGACGGTGGTATGTCTTTATG |
| Y275L278/T-F | This paper | CATAAAGACATACCACACTCTATATACAGCCATGTTAGAGAGGT |
| Y275L278/T-R | This paper | ACCTCTCTAACATGGCTGTATATAGAGTGTGGTATGTCTTTATG |
| F549R-F | This paper | GGCCCATCAGTGATAGGTTGACCTTGCACTCC |
| F549R-R | This paper | GGAGTGCAAGGTCAACCTATCACTGATGGGCC |
| F549S-F | This paper | GGCCCATCAGTGATTCTTTGACCTTGCACTCC |
| F549S-R | This paper | GGAGTGCAAGGTCAAAGAATCACTGATGGGCC |
| F549T-F | This paper | GGCCCATCAGTGATACTTTGACCTTGCACTCC |
| F549T-R | This paper | GGAGTGCAAGGTCAAAGTATCACTGATGGGCC |
| 546-553_del-f | This paper | CAGTACAACAGGCCCTCCATCATCATTAGC |
| 546-554_del-r | This paper | GCTAATGATGATGGAGGGCCTGTTGTACTG |
| Software and Algorithms | | |
| MotionCor2 | (Zheng et al, 2017) | https://msg.ucsf.edu/em/software/motioncor2.html |
| CTFFIND4 | (Rohou and Grigorieff, 2015) | https://grigorrielflab.umassmed.edu/software_download |
| CryoSPARC | (Punjani et al, 2017) | https://cryosparc.com/ |
| PyMOL | Schrödinger | https://pymol.org/2/ |
| EPU software | ThermoFisher Scientific | https://www.fei.com/software/epu-automated-single-particles-software-for-life-sciences/ |
| Chimera | (Pettersen et al, 2004) | https://www.cgl.ucsf.edu/chimera |
| ChimeraX | (Goddard et al, 2018) | https://www.cgl.ucsf.edu/chimerax/ |
| Coot | (Emsley et al, 2010) | https://www2.mrc-lmb.cam.ac.uk/personal/pemsley/coot |
| Phenix | (Adams et al, 2010) | https://www.phenix-online.org/ |
| PRISM 8.0 software | GraphPad | https://www.graphpad.com/scientific-software/prism/ |
| Illustrator | Adobe | https://www.adobe.com/cn/products/illustrator.html |
| DiscoverMP software | Refeyn Ltd, UK | https://refeyn.filecamp.com/l |
| PR.ThermoControl | NanoTemper | N/A |
| Creoptix WAVEcontrol software | Creoptix AG, Switzerland | N/A |
| TOPAZ 0.2.5a | (Bepler et al, 2019) | https://github.com/tbepler/topaz |
| DeepEMhancer | (Sanchez-Garcia et al, 2021) | https://github.com/rsanchezgarc/deepEMhancer |
| Other | | |
| SMM 293-TI | Sino Biological Inc | Cat. M293TI |
| FreeStyle 293 Expression Medium | ThermoFisher Scientific | Cat. # 12338026 |

**Table 4.** (continued)

| REAGENT or RESOURCE | SOURCE | IDENTIFIER |
|---|---|---|
| SIM SF | Sino Biological Inc | Cat. MSF1 |
| Ni NTA Beads 6FF | LABLEAD | N30210-100ml |
| Superdex200 Increase 10/300 GL | GE Healthcare | 28990944 |
| Superose6 Increase 10/300 GL | GE Healthcare | 29091596 |
| Resource Q | GE Healthcare | 17117901 |
| MagStrep "type3" Strep-Tactin® beads | IBA Lifesciences | 2-1613-002 |
| Amicon Ultra-15 10 KD | Millipore | UFC901096 |
| Slide-A-Lyzer™ G2 Dialysis Cassettes, 10K MWCO, 3 mL | Thermo Scientific | 87730 |
| UltraAuFoil grids | Quantifoil | N/A |

## Experiment model and subject details

DH5α strain of *E. coli* was used for molecular cloning and BL21(DE3) strain of *E. coli* was used for expression of recombinant FEM1B-ELOB-ELOC and its mutants. HEK293F cells obtained from Thermo Fisher Scientific were grown at 37 °C and 5% $CO_2$ in DMEM (Corning) supplemented with 1% fetal bovine serum (PAN Biotech), respectively. SF9 cells obtained from Invitrogen were grown at 27 °C in SIM-SF (Sino Biological Inc.).

## Molecular cloning and generation of constructs

All proteins are of human origin and are encoded by cDNAs amplified from a testis cDNA library. All the proteins and variants referred to were generated using PCR.

To overexpress FEM1B, ELOB, ELOC(17-112), BEX2, NAE1, UBA3, UBE2M, UBE2D3, UBE2R1, Ubiquitin and NEDD8(1–76) in *E. coli* BL21(DE3), constructs pET-28a(+)-6*HIS-MBP-TEV-FEM1B, pETDuet 1-ELOB(1-104)-ELOC(17-112), pET-28a(+)-6*HIS-MBP-BEX2, pETDuet 1-NAE1, pET-28a(+)-6*HIS-GST-TEV-UBA3, pET-28a(+)-6*HIS-TEV-UBE2M, pET-28a(+)-6*HIS-TEV-UBE2D3, pET-28a(+)-6*HIS-TEV-UBE2R1, pET-28a(+)-6*HIS-Ubiquitin and pET-28a(+)-6*HIS-TEV-NEDD8(1–76) were built. Genes encoding UBA1 and CUL2-RBX1 were cloned into a modified pFastBac Dual-6*His-MBP plasmid and pFastBac Dual to enable their expression in SF9 insect cells. The cDNAs encoding full-length FNIP1-FLCN were cloned into a pFastBac Dual vector, which had been modified to provide a CMV enhancer and promoter after an AcMNPV polyhedrin promoter and an EF1α promoter after a p10 promoter. FNIP1 cDNA was cloned with a mCherry-tag, and a 6*HIS-tag followed by 2*Strep-tag at its C-terminus.

## Protein expression

Expression vectors encoding the FEM1B/ELOB(1-104)/ELOC(17-112) complex, the UBA3/NAE1 complex, BEX2, UBE2M, UBE2D3, UBE2R1, Ubiquitin, and NEDD8 (1–76) were transformed into *E. coli* BL21(DE3) cells. Cells were grown in Luria-Bertani medium until reaching an OD600 0.6 and then induced with 0.1 mM IPTG overnight at 18 °C. Cells were harvested by centrifugation (4000 × *g* at 4 °C for 10 min) and flash frozen in liquid nitrogen.

The recombinant baculoviruses encoding UBA1 and the CUL2/RBX1 complex were amplified in SF9 insect cells according to the manufacturer's instructions. After 60 h of suspension culture at 27 °C, cell pellets were collected by centrifugation and flash frozen as before.

The FNIP1/FLCN complex was expressed using the BacMam system. The modified baculoviruses of FNIP1/FLCN complex were prepared as above. HEK293F cells grown in suspension at 37 °C (6% $CO_2$, 70–80% humidity) were infected with 10% baculoviruses, and after 12 h, 10 mM sodium butyrate was added. After another 48 h, pellets were collected, and flash frozen in liquid nitrogen and stored at −80 °C.

## Protein purification

For proteins expressed in *E. coli* BL21(DE3) cells (800 ml LB culture) and SF9 insect cells (500 ml cell culture), cell pellets were resuspended in lysis buffer containing 30 mM Tris-HCl pH 7.5, 140 mM NaCl, 3 mM KCl, 10% glycerol, 20 mM imidazole, 1 mM phenylmethylsulfonyl fluoride (PMSF) and 0.5 mM TCEP, followed by sonication on ice. After centrifugation at 45,000 × *g* for 30 min, the supernatant was collected and filtered through cheesecloth. A 10 ml Ni-NTA column equilibrated with lysis buffer was incubated with filtered supernatant for 1 h at 4 °C and then washed with buffer A (30 mM Tris-HCl pH 7.5, 500 mM NaCl, 20 mM imidazole, and 1 mM TCEP). Recombinant proteins were eluted with buffer C (30 mM Tris-HCl pH 7.5, 500 mM NaCl, 300 mM imidazole, and 1 mM TCEP) and tags were removed by incubation with TEV protease overnight. The protein solution was concentrated and loaded onto a Superdex 200 increase 10/300 gl column (GE Healthcare) equilibrated with 30 mM Tris-HCl pH 7.5, 150 mM NaCl, and 1 mM TCEP. Purified proteins were concentrated, aliquoted, and flash frozen using liquid nitrogen, yielding ~5 mg of MBP-FEM1B-ELOB-ELOC and ~1 mg of CUL2-RBX1 per run.

The purification of the complete CUL2/RBX1/ELOB/ELOC/FEM1B (CRL2^FEM1B) ligase complex was performed in manner similar to that described above. Purified CUL2/RBX1 complex and FEM1B/ELOB(1-104)/ELOC(17-112) complex were mixed at 1:1 molar ratio on ice. The mixture was dialyzed against 50 mM HEPES, pH 7.5, 100 mM NaCl, and 5 mM DTT at 4 °C overnight and TEV protease was added to cleave the MBP tag from FEM1B. Filtered reactions were loaded onto a 6 ml Resource Q IEX column (Cytiva) equilibrated with buffer Wa (50 mM HEPES pH 7.5, 100 mM NaCl, and 5 mM DTT) and washed with buffer Wa until

the absorbance had returned to baseline. The CRL2$^{FEM1B}$ ligase complex was eluted using a linear gradient ranging from 150 mM to 300 mM NaCl with buffer Wb (50 mM HEPES pH 7.5, 1 M NaCl, and 5 mM DTT). Fractions of cleaved CRL2$^{FEM1B}$ ligase complex were concentrated and further purified using a Superose 6 increase 10/300 gl column (GE) in 50 mM HEPES pH 7.5, 150 mM NaCl, and 1 mM TCEP.

HEK293F cells (400 ml cell culture) expressing the FNIP1/FLCN complex were resuspended in lysis buffer (50 mM HEPES pH 7.5, 140 mM NaCl, 3 mM KCl, 10% glycerol, 1 mM PMSF, 1 mM TCEP, and 20 mM imidazole) and sonicated on ice. The supernatant was collected and filtered as described above after centrifuging twice at $45,000 \times g$ for 30 min each time. A 10 ml Ni-NTA column was equilibrated with buffer W (50 mM HEPES pH 7.5, 150 mM NaCl, 20 mM imidazole, and 1 mM TCEP) and incubated with filtered supernatant for 1 h at 4 °C. After incubation, beads were washed with buffer W until the absorbance had returned to baseline and then the proteins eluted with Buffer E (50 mM HEPES pH 7.5, 150 mM NaCl, 300 mM imidazole, and 1 mM TCEP). Eluted proteins were loaded onto a 5 ml Strep Trap column and tags were removed using TEV protease. The flow-through was collected and concentrated by Amicon Ultra-15 10KD filters before further purification of the FNIP1/FLCN complex using a Superose 6 increase 10/300 gl column (GE Healthcare) equilibrated with 50 mM HEPES pH 7.5, 150 mM NaCl, and 1 mM TCEP. Fractions containing FNIP1/FLCN complex were collected, concentrated, aliquoted, and flash frozen using liquid nitrogen.

### In vitro neddylation

Neddylation of all variants was performed in a reaction mixture containing 50 mM HEPES pH 7.5, 150 mM NaCl, 10 mM MgCl$_2$, 1 mM DTT, 20 mM ATP, 6.3 μM NEDD8, 5 μM CRL2$^{FEM1B}$, 700 nM UBA3, and 2 μM UBE2M for 4 h at room temperature. Reactions were terminated by quenching into 10 mM DTT.

### In vitro ubiquitylation

For in vitro ubiquitylation, all variants of CRL2$^{FEM1B}$ ligase complex were first activated by neddylation as described above. Ubiquitylation was carried out in 50 mM HEPES pH 7.5, 100 mM NaCl, 10 mM MgCl$_2$, 50 μM ZnCl$_2$, 1 mM DTT, 2 mM ATP, 1 μM neddylated CRL2$^{FEM1B}$ ligase, 10 μM ubiquitin, 1 μM UBE2D3, 1 μM UBA1, and 1 μM substrate at room temperature and the reactions terminated with 10 mM DTT (Manford et al, 2020).

### Western blotting

For western blotting, samples were first separated on a 4–20% Precast Protein Plus Gel (15 wells, Hepes-Tris) and then transferred to polyvinylidene difluoride membranes under conditions of constant current. The membranes were blocked with 5% (w/v) non-fat milk in Tris-buffered saline (TBS) containing 0.1% Tween 20 (TBST) for 1 h at room temperature and probed with primary antibody [anti-FEM1B (Abclonal, A12802; RRID: AB_2759641), anti-FNIP1 (Abcam, ab134969), anti-UBA1 (Abcam, ab181225), anti-UBE2R1 (Abcam, ab204515), anti-UBE2D3 (Abcam, ab176568) or anti-MBP (Abcam, ab119994; RRID: AB_10900967)] overnight at 4 °C. Subsequently, membranes

were washed in TBST and further probed with goat anti-rabbit (Pierce, 31460) conjugated secondary antibodies. Immunoreactive bands were visualized with the western blotting luminol reagent (Santa Cruz Biotechnology Inc., sc-2048).

### Size-exclusion chromatography multi-angle light scattering

A high-pressure injection system (Wyatt Technology) and a chromatography system equipped with a DAWN HELEOS-II MALS detector, and an Optilab T-rEX differential refractive index detector was used to perform SEC-MALS analysis. An aliquot of 100 μl protein at 1 mg/ml was loaded onto a WTC-015S5 column (7.8 × 300 mm, 5 μm, Wyatt Technology) and eluted at a flow rate of 0.4 ml/min in 50 mM HEPES, pH 7.5, 150 mM NaCl, and 1 mM TCEP. The outputs were analyzed, and molecular masses determined using the Astra 6 software program (Wyatt Technology) from the Rayleigh ratio calculated by measuring the static light scattering and corresponding protein concentration of a selected peak.

### Negative staining electron microscopy

The protein samples were prepared by diluting them to 0.03 mg/ml and spinning them at 12,000 rpm for 10 min at 4 °C. The carbon films were plasma-cleaned for one minute at low setting. Then, 5 μl of protein was pipetted onto the film and left for one minute before blotting with filter paper. The film was rinsed with water and 1% uranium acetate for 10 s each and stained with 1% uranium acetate for one minute before blotting again. The negatively stained grids were examined with a JEM-1400PLus electron microscope at 100 kV and an EMSIS CCD camera at 40,000× magnification.

### Mass photometry

NISTmAb (humanized IgG1 monoclonal antibody, RM 8671) and thyroglobulin (T9145) were acquired from National Institute of Standards and Technology (USA) and Merck, respectively.

Mass photometry (MP) experiments were performed using a Two$^{MP}$ mass photometer (Refeyn Ltd, UK). MP movies were recorded at a frequency of 475.2 Hz, with exposure times set at 2.06 ms. These settings were carefully adjusted to optimize camera counts while preventing saturation. Prior to the measurements, the instrument was calibrated using NISTmAb (148 kDa) and thyroglobulin (660 kDa). To establish focus, a fresh buffer (50 mM HEPES pH 7.5, 150 mM NaCl) was pipetted into a well in an 18 μL volume. The focal position was identified and locked using the instrument's autofocus function. For each acquisition, 2 μL of protein solution at a concentration of 10 nM was added to the well. The acquired data were analyzed using the DiscoverMP software (v2023 R2).

### Cryo-EM sample preparation

Unneddylated CRL2$^{FEM1B}$ ligase complexes were further purified and crosslinked using the GraFix method (Kastner et al, 2008) at 33000 rpm for 18 h. Glycerol was removed from the stabilized samples by dialyzing against 50 mM HEPES, 100 mM NaCl, and 5 mM DTT, and concentrated to 1.42 mg/ml.

For neddylated CRL2$^{FEM1B}$-BEX2 complex, neddylated CRL2$^{FEM1B}$ and BEX2 were mixed in a 1:3 molar ratio with 50 μM ZnCl$_2$ and

incubated for 1 h on ice. As for neddylated CRL2$^{FEM1B}$-FNIP1-FLCN samples, neddylated CRL2$^{FEM1B}$ and FNIP1-FLCN complexes were mixed in a 1:1.2 molar ratio and incubated for 1 h at 4 °C. After GraFix (Kastner et al, 2008), dialysis, and concentration, stabilized samples at a concentration of 1 mg/ml and 1.38 mg/ml were obtained, respectively.

Samples (3 µl) were applied to glow-discharged UltraAuFoil holey gold grids (Quantifoil R 0.6/1.0, Au 300) and plunge-frozen by a Vitrobot Mark IV (Thermo Fisher) in liquid ethane chilled with liquid nitrogen with the following settings: blot force 1, blot time 1, wait time 15 s, 100% humidity, and 4 °C.

## Data collection

Grids were screened at Peking University using a 200 kV Talos Arctica Cryo-TEM (Thermo Fisher) and a 300 kV Titan Krios microscope (Thermo Fisher) with a K2 direct electron detector (Gatan). High-resolution data collections were performed at Southern University of Science and Technology and Shuimu BioSciences on a 300 kV Titan Krios (Thermo Fisher) microscope with a K3 direct electron detector or a Falcon 4 direct electron detector, respectively. Datasets were collected in counting mode at 105,000× and 96,000× magnifications with calibrated pixel sizes of 0.83 Å and 0.86 Å, respectively. The dose rate was set to approximately 25 e-/pixel/s with an exposure time of 1.36 s (total dose was 50 e/Å$^2$). 5400 images were collected for the unneddylated CRL2$^{FEM1B}$ ligase complex, with a defocus range of −1.5 µm to −2.0 µm. 5989 images were collected for the neddylated CRL2$^{FEM1B}$-BEX2 complex contains while 15,000 images were collected for neddylated CRL2$^{FEM1B}$-FNIP1-FLCN complex with the same defocus range.

## Data processing

Cryo-EM datasets were processed using cryoSPARC v4.2.1 (Punjani et al, 2017; Rohou and Grigorieff, 2015; Rubinstein and Brubaker, 2015; Stagg et al, 2014; Zheng et al, 2017). Image stacks of unneddylated CRL2$^{FEM1B}$ and neddylated CRL2$^{FEM1B}$-FNIP1-FLCN complex were aligned using Patch motion correction, and defocus value estimation was performed by Patch CTF estimation. Image stacks of neddylated CRL2$^{FEM1B}$-BEX2 were aligned using MotionCor2 and defocus value estimation was performed by Patch CTF estimation. Manually selected particles were used to create templates for template-based picking. After several iterations of 2D Classification, particles from the highest-resolution classes were used for 3D ab initio reconstruction. Heterogeneous refinement was performed to pick the best 3D class, followed by homogeneous refinement and non-uniform refinement (Punjani et al, 2020) to improve the resolution. Best particles were then used as template for TOPAZ (v0.2.5a) particle picking (Bepler et al, 2019). After multiple rounds of heterogeneous refinement, best particles were used for homogeneous refinement and non-uniform refinement to improve the resolution.

## Cryo-EM density map interpretation and model building

Maps were sharpened by B-factor applied and DeepEMhancer (Sanchez-Garcia et al, 2021) before model building. The atomic coordinates were modeled and refined according to amino acid main chain and side chain densities. Known models of components were fitted with rigid-body refinement using UCSF Chimera (Pettersen et al, 2004) and UCSF ChimeraX (Goddard et al, 2018). Manual model building was performed in Coot (Emsley and Cowtan, 2004) and refined in Phenix (Adams et al, 2010; Liebschner et al, 2019).

## Glycerol gradient ultracentrifugation and GraFix

For gradient ultracentrifugation, two buffer solutions were prepared, each containing 50 mM HEPES, pH 7.5, 50 µM ZnCl$_2$, and 100 mM NaCl, together with 10% (v/v) glycerol for the top buffer and 30% (v/v) glycerol for the bottom buffer. The pre-filtered (0.22 µm) buffers were layered in a 5 ml polypropylene centrifuge tube (326819, Beckman Coulter) by carefully adding 2.5 ml of the top buffer on top of the bottom buffer. After standing for 1 h at 4 °C to allow a continuous density gradient to form, purified protein sample (200 µl at 1 mg/ml) was loaded on top and tubes balanced prior to placing into a pre-cooled (4 °C) rotor (MLS-50, Beckman Coulter). Ultracentrifugation was carried out at 4 °C for 18 h at 33,000 rpm and fractions (200 µl) were collected at 4 °C from the top of the gradient. For GraFix, we followed the same protocols as for glycerol gradient ultracentrifugation, except that we added 0.15% (v/v) GA to the bottom buffer. The bottom buffer consisted of 50 mM HEPES (pH 7.5), 50 µM ZnCl$_2$, 100 mM NaCl, 30% (v/v) glycerol, and 0.15% (v/v) GA.

## NanoDSF measurements

NanoDSF measurements were performed on a Prometheus NT.48 (NanoTemper) using the corresponding software for data collection and analysis. Proteins were adjusted to 0.5 mg/ml using buffer HBS (20 mM HEPES pH 7.5, 150 mM NaCl). Standard capillaries (NanoTemper, Cat# PR-C002) were filled with protein solution and directly loaded into the device. The temperature range for measurements was set to 20–90 °C with a temperature ramp of 1 °C per minute. All measurements were performed in three technical replicates.

## Grating-coupled interferometry

The Creoptix WAVE system (Creoptix AG, Switzerland), a label-free surface biosensor was used to perform GCI experiments. All experiments were performed on 4PCP WAVEchips (quasi-planar polycarboxylate surface; Creoptix AG, Switzerland). After a borate buffer conditioning (100 mM sodium borate pH 9.0, 1 M NaCl; Xantec, Germany), the MBP-BEX2 and FNIP1/FLCN were immobilized on the chip surface using standard amine-coupling: 7 min activation (1:1 mix of 400 mM N-(3-dimethylaminopropyl)-N'-ethylcarbodiimide hydrochloride and 100 mM N-hydroxysuccinimide (both Xantec, Germany)), injection of MBP-BEX2 and FNIP1/FLCN (10 µg/ml) in 10 mM sodium acetate pH 4.5 (Cytiva, Sweden) until the desired density was reached, passivation of the surface and final quenching with 1 M ethanolamine pH 8.0 for 7 min (Xantec, Germany).

For a typical experiment, CRL2$^{FEM1B}$ was injected in a 1:3 dilution series (starting from 1 µM) in 10 mM HEPES pH 7.5, 150 mM NaCl, 50 µM ZnCl$_2$ at 25 °C. Blank injections were used for double referencing. Analysis and correction of the obtained data

was performed using the Creoptix WAVEcontrol software (applied corrections, X and Y offset; DMSO calibration; double referencing) and a 1:1 kinetics binding model with bulk correction was used to fit all experiments.

## Data availability

The cryo-EM map and atomic coordinates of the dimeric CRL2$^{FEM1B}$ complex have been deposited in the EMDB and PDB under ID codes EMD-35461 and PDB: 8IJ1, respectively. The accession numbers for the neddylated CRL2$^{FEM1B}$-BEX2 cryo-EM map and coordinates reported in this paper are EMD-36182 and PDB: 8JE1, respectively. Cryo-EM data of neddylated CRL2$^{FEM1B}$ complexed with FNIP1-FLCN have been deposited at PDB and EMDB under ID codes PDB: 8JE2 and EMD-36183, respectively.

## Peer review information

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

## Acknowledgements

We thank the cryo-EM facility of Southern University of Science and Technology (Shenzhen, China) and Shuimu BioSciences (Beijing, China) for providing support in cryo-EM data collection; Dr. Xuemei Li from the Electron Microscopy Laboratory (Peking University) for providing facility support in cryo-EM sample screening; Dr. Qian Wang and Dr. Jing Wang from the State Key Laboratory of Natural and Biomimetic Drugs (Peking University, Beijing, China) for assistance with the NanoDSF measurements; Dr. Lei Zhang from Peking University Medical and Health Analysis Center for negative staining

electron microscopy data collection; Dr. Peiwei Han and Mrs. Yi Qin from Malvern Panalytical for providing support in Grating-coupled interferometry; Dr. Rongrong Dai (Mass Spectrometry Core, Changping Laboratory, China) for assistance with the mass photometry measurements. Dr. Lin Yuan (Peking University Shenzhen Hospital), Dr. Haikel Dridi and Dr. Oliver B. Clarke (Columbia University in the City of New York, Vagelos College of Physicians & Surgeons) for discussions and comments; Dr. Susan P.C. Cole from Queen's University Cancer Research Institute (Kingston, Canada) for review and editing of this manuscript. This study is supported by grants to Y. Yin including National Natural Science Foundation of China (key grants 82030081 and and 81874235), the National Key Research and Development Program of China (2021YFA1300601), The Shenzhen High-level Hospital Construction Fund and Shenzhen Basic Research Key Project (JCYJ20220818102811024). Y. Yin is the Scholar of the Lam Chung Nin Foundation for Systems Biomedicine.

## Author contributions

**Zonglin Dai**: Investigation; Methodology; Writing—original draft; Writing—review and editing. **Ling Liang**: Investigation; Writing—review and editing. **Weize Wang**: Investigation. **Peng Zuo**: Investigation. **Shang Yu**: Investigation. **Yaqi Liu**: Writing—review and editing. **Xuyang Zhao**: Investigation. **Yishuo Lu**: Investigation. **Yan Jin**: Supervision. **Fangting Zhang**: Resources. **Dian Ding**: Investigation. **Weiwei Deng**: Resources. **Yuxin Yin**: Conceptualization; Supervision; Funding acquisition; Writing—review and editing.

## Disclosure and competing interests statement

The authors declare no competing interests.

