## [Peer Review File · The EMBO Journal]

Structural Insights into the Ubiquitylation Strategy of the Oligomeric CRL2FEM1B E3 Ubiquitin Ligase

Zonglin Dai, Ling Liang, Weize Wang, Peng Zuo, Shang Yu, Yaqi Liu, Xuyang Zhao, Yishuo Lu, Yan Jin, Fangting Zhang, Dian Ding, Weiwei Deng, and Yuxin Yin

Corresponding author(s): Yuxin Yin (yinyuxin@hsc.pku.edu.cn)

Review Timeline:

Submission Date:	22nd Aug 23
Editorial Decision:	29th Sep 23
Revision Received:	12th Dec 23
Editorial Decision:	23rd Jan 24
Revision Received:	26th Jan 24
Accepted:	29th Jan 24

Editor: Hartmut Vodermaier

Transaction Report:

Prof. Yuxin Yin
Peking University Health Science Center
Institute of Systems Biomedicine
38 Xue Yuan Road
Beijing 100191
China

29th Sep 2023

Re: EMBOJ-2023-115372
Structural Insights into the Ubiquitylation Strategy of the Oligomeric CRL2FEM1B E3 Ubiquitin Ligase

Dear Dr. Yin,

Thank you again for submitting your manuscript on oligomeric CRL2FEM1B ligase structures to The EMBO Journal. It has now been reviewed by three expert referees, whose reports are copied below. The referees appreciate the importance of the subject and the potential interest of your new findings. However, they all raise a number of significant concerns that would need to be addressed before publication would be warranted. These issues mostly concern missing method details of the structural and biophysical analyses, and insufficient use of cryo-EM processing approaches. There are also concerns about mutation verification and additional/better mutations being needed to validate structures. Finally, the referees mandate better presentations both in the text and the figures.

Should you be able to satisfactorily address the criticisms by all three referees, we would be interested in considering a revised manuscript further for publication. Since it is our policy to only allow a single round of major revision, it will however be important to fully and carefully respond to each referee point at the time of resubmission. I would therefore encourage you to contact me with a revision plan and preliminary point-by-point response already during the early stages of your revision work, in order to clarify if and how key issues raised in the reports may be solved. We might also allow extension of the default three-months revision period if needed; our 'scooping protection' (meaning that competing work appearing elsewhere in the meantime will not affect our considerations of your study) would of course remain valid also throughout such an extension.

Further information on preparing, formatting and uploading a revised manuscript can be found below and in our Guide to Authors. Thank you again for the opportunity to consider this work for The EMBO Journal, and I look forward to hearing from you in due time.

Yours sincerely,

Hartmut Vodermaier

9) Digital image enhancement is acceptable practice, as long as it accurately represents the original data and conforms to community standards. If a figure has been subjected to significant electronic manipulation, this must be clearly noted in the figure legend and/or the 'Materials and Methods' section. The editors reserve the right to request original versions of figures and the original images that were used to assemble the figure. Finally, we generally encourage uploading of numerical as well as gel/blot image source data; for details see: embopress.org/page/journal/14602075/authorguide#sourcedata

At EMBO Press, we ask authors to provide source data for the main manuscript figures. Our source data coordinator will contact you to discuss which figure panels we would need source data for and will also provide you with helpful tips on how to upload and organize the files.

Revision to The EMBO Journal should be submitted online within 90 days, unless an extension has been requested and approved by the editor; please click on the link below to submit the revision online before 28th Dec 2023:
Link Not Available

If you choose to alternatively have this study further considered by another EMBO Press publication, please use the following hyperlink to directly transfer the manuscript, optionally with inclusion of referee reports and identities:
Link Not Available

Referee #1:

Dai et al. offer a structural investigation into the oligomerization of Cullin-RING Ligases (CRLs), highlighting potential functional roles in substrate recognition. Utilizing cryo-electron microscopy (cryo-EM), the study identifies both dimeric and trimeric states of these ligases. Although the subject matter is compelling, the manuscript and the corresponding structural data exhibit several significant shortcomings that must be addressed prior to publication.

1. Clarity and Structure: The manuscript is notably difficult to follow, owing to the authors' brevity in explaining complex protein complexes. To enhance readability, the manuscript requires streamlining, and the individual complexes warrant better rationalization. The use of cartoon representations could replace lengthy, confusing nomenclature and make the content more accessible to readers not deeply versed in the CRL field. For instance, the second paragraph of the introduction abruptly introduces numerous complexes and facts, making it challenging for readers to understand the logical connections to the study's core focus in the third paragraph. However, this comment refers not only to these exemplified sections but the entire manuscript which lacks description of results as well as rationals for individual approaches.

2. Quality of Structures: The resolved structures, as presented, are of low quality-appearing smeary and poorly resolved. This is likely due to the simplistic data processing methods and the insufficient number of particles employed. Advanced computational tools like 3DFlex or DeepEMhancer, or a more meticulous 3D classification, could enhance the analysis. While the flexibility of CRL complexes is acknowledged, their oligomeric states appear more stable and deserve better analytical rigor. Also the models are not well done. Many side chains are actually out of density while the density would be available. This is especially critical in the interfaces. If amino acids should be interpreted, as the authors do, the fits have to be reasonable. I am also lacking the customary "Table 1" detailing the acquisition, image processing and modelling including its quality control. This can not be missed in any cryoEM paper and should be included.

3. Mutation Verification: The mutations employed for interface validation seem drastic. For instance, changing Leucine or Phenylalanine to Arginine results in a significant shift from hydrophobic to charged properties. A subtler mutation like Leucine to Serine or Threonine could offer a more convincing argument. There is also a lack of quality control to confirm that these mutations don't alter the overall protein fold. The inclusion of at least one stability assay is essential for demonstrating that the mutations specifically disrupt the binding interface. Quantitative analyses, such as affinity measurements, are also desirable.

4. Oligomeric State Assessment: The mass photometry data reveal only minimal amounts of oligomers, raising questions about their actual functional significance. Especially when going to assays. Also, the native page does not provide clearer evidence. Alternative methods like SEC-MALS might provide a more robust evaluation of oligomeric states in these complexes. Additionally, the mass photometry seems to have issues judging by the negative peaks in the assay. When using well prepared sample and surface these artifacts should not occur.

5. Ubiquitylation Assay Description: The ubiquitylation assay is pivotal to the study but is poorly described. Additional details, especially relating to critical figures like Figure 5B, are needed. Visual aids illustrating the constructs used could improve understanding. Again, it is vital to be quantitative and perform assays that can give physical parameters rather than just an endpoint visualisation.

6. The manuscript refers to Figures S3 and S5 when discussing the cryo-EM structure of the N8-CRL2FEM1B-BEX2 complex. These figures appear to detail the workflow rather than the actual structure, warranting clarification from the authors.

7. Methodological Details: The Grafix procedure described in the methods section lacks critical details such as the type of gradient, rotor, and GA concentration used, which could affect replicability and validation of the results.

Minor Points:

8. The term "E1 ubiquitin ligase" is incorrectly used in the first paragraph; E1 refers to the ubiquitin-activating enzyme and not a ligase.

9. The current citation style hinders readability by placing all citations at the end of sentences. Placing each citation immediately after the relevant claim would improve clarity.

10. Grammatical issues, including missing articles, should be rectified for a polished, professional manuscript.

In summary, while the research presents an intriguing focus on the structural complexities of Cullin-RING Ligases, there are several major and minor issues that need to be addressed to make the manuscript suitable for publication.

Referee #2:

Review of Structural Insights into the Ubiquitylation Strategy of the Oligomeric CRL2FEM1B E3 Ubiquitin ligase

Substrate recognition mechanism by the ankyrin-repeat domain of FEM1B, a substrate adapter for Cul2 ring ligases, has been described previously (Chen, 2021, Manford, 2021 and others) through structures of the N-terminal region of FEM1B bound to C-Arg degrons and FNIP1 substrates.

To understand the regulation and oligomerization state of an intact FEM1B-E3 complex, the authors report several cryo-EM structures of reconstituted CRL2-Rbx1-ElonginB/C-Fem1B complexes.

The authors report different structural arrangements of Cul2Fem1B in the un-neddylated state versus neddylated state determined with BEX2 and FNIP1/FLCN substrates.

CRL2FEM1B forms a dimer and trimeric state.

N8- CRL2FEM1B-MBP-Bex2 complex forms a dimer related to above (slight movement of FEM1B substrate pocket of protamer 1 and altered positioning of catalytic Rbx1 protamer 2)

N8- CRL2FEM1B -FNIP1/FLCN forms a monomer.

The structural biology aspect of this paper is a tour du force and major strength.

However, the interpretation of the structures, in particular the significance of the asymmetric oligomeric E3 complex (dimeric, trimeric and monomeric versions) and their functional validation could be strengthened.

The authors claim the asymmetric dimer configuration forms a composite functional complex with the substrate recruitment and catalytic Ub transfer undertaken by distinct protamer subunits in the Cul2FEM1B dimer. This dimeric arrangement is not required or observed for complexes with a large/intact substrate FNIP1/FLCN.

To test the significance of the observed super-complex interfaces, the authors generated three mutant versions of FEM1B. and tested them using a mixture of oligomerization, substrate binding and ubiquitination assays.

Mass photometry was used to assess oligomerization of Cul2Fem1B and mutants. Additional data from multi-angle light scattering, ultracentrifugation and direct protein visualization of bands on native gels is reported for select samples.

Unfortunately, these above techniques were not uniformly applied to all complexes which makes it difficult to assess the validity of their conclusions.

For example, Fem1B interface mutants F549R, Y277R/L278R and a loop deletion (DEL1 546-553) disrupt the trimeric state completely and dimeric form partially? (DEL1 mutant only) in the mass photometry assay (Fig Supp.8).

Functionally, only the DEL1 and F549R mutant were tested and shown to retain FNIP1 binding (Fig.S9).

These mutants were only tested for ubiquitination of the MBP-BEX substrate (Fig 6G) but not the FNIP1/FLCN substrate. Is the monomeric DEL1 mutant still functional to ubiquitinate FNIP1/FLCN?

The role of Cul2-Neddylation on the oligomeric state was also examined. A neddylated-Cul2Fem1B complex was able to ubiquitinate each substrate examined (although the unneddylated version was not tested). No trimeric version of the N8-Cul2FEM1B complex was observed by mass photometry.

Further evidence of the impact Neddylation has on complex formation was examined by a cryo-EM structure of Neddylated Crl2FEM1B in complex with MBP-Bex2. Given the Neddylation site (lys689) is at a Cul2-Cul2 interface of the unneddylated-oligomeric complex (Fig.4a), a description of the structural changes that occur when this site on Cul2 is Neddylated was lacking.

Finally, a structure was determined in the presence of FNIP1/FLCN. The extent of map contouring to support substrate engagement was not convincing to this reviewer. Perhaps better datasets will provide more evidence to support a full Cul2FEM1b-FNIP1/FLCN complex structure. The authors state that monomeric Cul2FEM1B is capable of ubiquitinating this intact protein substrate yet there is no functional data demonstrating this with a monomeric-interface mutant such as the DEL1.

Other points to address:

-How can uniform Neddylation on K687 of Cul2 be consistent with dimerization given the view provided in Figure 4? This structural hypothesis appears inconsistent with complete Cul2-Neddylation demonstrated in Fig 4B yet a prominent dimeric band on a native gel in Fig 4C.

-Authors claim that dimeric Cul2FEM1B is active for ubiquitination of MBP-Bex and MBP-FNIP1-degron substrates while Cul2FEM1B monomer ubiquitinates FNIP1/FLCN. Oligomeric states of these Cul2FEM1B-substrate complexes are compared by different methods however (Blue native PAGE versus SEC-MALS and mass photometry respectively). Can the authors demonstrate the dimeric nature of the BEX or FNIP1-degron complex by SEC-MALS and or mass photometry for a direct comparison? Does a monomeric Cul2FEM1B complex ubiquitinate FNIP1/FLCN since this was only attempted for the MBP-Bex substrate.

Minor points:

-EM map not clearly visible in Figure 6E.

-Oligomeric interface mutants investigated: Y277R/L278R - only L278 is visible in the interface shown in Figure 3B/Fig S7B. Why was Y277 targeted for mutagenesis since this sidechain does not appear oriented towards protamer 2 (while Y275 appears a better candidate on this basis). Were any other interface mutants tested and not shown in the manuscript?

-any of the cross links from Gradient Fixation visible in the structures? Was any structural classes obtained from non Grafix datasets that are consistent with the oligomeric super-complexes described?

Referee #3:

Summary

The three equal contribution authors Dai, Liang, and Wang in this "Dai et al." study have applied single particle cryo-EM to the cullin-RING E3 ligase, Cul2/FEM1B (CRL2FEM1B). This resulted in three reported structures: CRL2FEM1B unneddylated, CRL2FEM1B/N8 + BEX2, and CRL2FEM1B/N8 + FNIP1/FLCN. Providing insight to how CRL2FEM1B oligomerizes, their structures reveal an asymmetric dimer as well as a trimeric form. These findings are supported by several biophysical methods: SEC-MALS, mass photometry, and native PAGE. CRLs are inherently flexible and achieving near-atomic resolution has proved futile. Many excellent papers have applied single particle cryo-EM to CRLs and CUL2 based complexes resulting in resolution from ~12 - 3.5 Å. The work of the Dai et al. study builds on our need to understand CRLs as complete complexes. Their Neddylation of CRL2FEM1B enabled an active E3 ligase that had activity in several assays. Importantly, point mutations at the promoter interfaces reduced the oligomeric state and this was also reflected in the in vitro ubiquitination assays. Overall, this Dai et al. study reports novel structures of oligomerized CRL2FEM1B and supporting biophysical work.

Some wording, especially in the introduction make it difficult to follow. The methods section needs to be expanded on to provide the reader a clear idea on how key experiments in the main figures and supplementary figures were carried out. In general many key experimental details are missing (documented below). The use of the term, "the first..." does come up frequently and could be reduced. With the recent developments in cryo-EM, could the 3D heterogeneous flexible refinement e.g. Zhong 2021 CryoDRGN (<https://doi.org/10.1038/s41592-020-01049-4>) be helpful. There 3D classes and 3D variability were mentioned, but only presented as volumes. In addition, could cryo-EM map enhancement (e.g. DeepEMhancer Sanchez-Garcia et al. 2021) <https://doi.org/10.1038/s42003-021-02399-1> be helpful in further resolving RBX1, FEM1B, and other parts ?

I could accept this study with many of the minor revisions.

Minor Comments

- Abstract => be careful using the term "the first" Maniaci 2017 (<https://doi.org/10.1038/s41467-017-00954-1>) have induced oligomerization for CRL2VHL and since then other homo- and hetero-PROTACs have been reported. Your introduction goes on to cite numerous other examples.
- Introduction => first sentence "Ring" is usually "RING" for E3 ubiquitin ligases (Really Interesting New Gene) with RBX1 and RBX2 being the only known for CRLs.
- Introduction => "In the proteasomal degradation system or ubiquitin- proteasome system (UPS)" is repetitive.
- Introduction => "poly-ubiquitin" can be one word "polyubiquitin" or "polyUb"
- Introduction => "E2 ubiquitin ligases" this is usually "E2 ubiquitin-conjugating enzyme (E2) and E3s are the ligases."
- Introduction => "E1 ubiquitin ligases (E1s)" usually E1 refers to ubiquitin activating enzyme (E1)
- Introduction => "It has also been reported that CUL2 could not only assemble hetero-dimeric CRLs with CUL4A" this wording is confusing. "It has also been reported that CUL2 assembles hetero-dimeric CRLs with CUL4A (which is involved in the progression of Alzheimer's disease) (Yasukawa et al, 2020), in addition to forming homo-dimeric CRL2VHL complex in vivo (Chung et al, 2006; Merlet et al, 2009)."
- Introduction => "these studies provide strong evidence that oligomerization of CRLs occurs not only via substrate recognition proteins but also via adaptors, or Cullin scaffolds, or their combinations as well (Bulatov & Ciulli, 2015)." Change to "these studies provide strong evidence that oligomerization of CRLs occurs not only via substrate recognition proteins but also via adaptors, Cullin scaffolds, or their combinations (Bulatov & Ciulli, 2015)."
- Results => "by His-Trapping and Q-Column" this could just be "IMAC and anion exchange" The use of "His-Trapping" is not ideal here.
- Results => "Using standard cryo-EM methods, we solved the structure of purified recombinant trimeric CRL2FEM1B to 7.38 Å resolution" by "standard" do you mean conventional single particle ?
- Results => "Cryo-EM structure of CRL2FEM1B dimeric and trimeric super-complex" This section read like a methods section in parts, but regardless should end with a concluding sentence.
- Results => "When the two structures are superimposed, the CUL2-RBX1 scaffolds and adaptor proteins differ slightly, with a root-mean-square deviation of 2.088 Å (SI Appendix, Fig. S6A)" is this RMSD alpha carbons (Cα) or something else
- Results => "Furthermore, the loop connecting -helix D and -helix E of the CUL2 cullin repeat 1 (which was not visible in previously reported structures (Cardote et al., 2017)), appears to bind the C-terminus of FEM1B through hydrophobic interactions" you can re-word to include "which was not resolved in the previously reported X-ray crystal structure (Cardote et al., 2017)" There seems to have only been on structure reported PDB-5N4W.
- Discussion => there are great examples of oligomerizing substrate adapters and CRLs. However, given your focus on the C-terminus and RBX1 would it make sense to expand on how E2~Ub or E2~N8 binding to RBX1 in CRL2FEM1B could alter the oligomer ?

- Material and Methods => Experiment model and subject details "E. coli were used for molecular cloning and expression of recombinant FEM1B-ELOB-ELOC and its mutants." Did you use two different strains for E. coli. It is highly important to report the expression strain (e.g. BL-21 Rosetta2).
- Materials and Methods => Molecular Cloning and Generation of Constructs "All proteins are of human origin and are encoded by cDNAs amplified from a testis cDNA library. All the proteins and variants referred to were generated using PCR." Were regular BL-21 (DE3) cells used ? If so were there any issues with codons when using human cDNA and an E. coli strain that may lack some of the tRNAs for rare codons ?
- After reading the Materials and Methods, why is the first section there, it is repeated and expanded on in the following sections.
- Materials and Methods => Protein purification what scale are we talking about ~1mg, or tens of mg of protein. In general try to also include the size/total column volume of the columns used (e.g. 5 ml His-Trap, 10/300 Superose6, etc).
- Materials and Methods => Protein purification "The CRL2FEM1B ligase complex was eluted using a linear gradient ranging from 150 mM to 300 mM NaCl with buffer Wb (50 mM HEPES pH 7.5, 1 M NaCl and 5 mM DTT)." For how many column volumes was this gradient and at what % did the CRL elute ?
- Materials and Methods => Western Blotting | can you describe the type of gel used (e.g. 10% SDS-PAGE, 1mm, Bis-Tris, Life Technologies) and also the running buffer (Tris-Glycine, MES, MOPS, etc).
- Materials and Methods => Mass Photometry | Include the final concentration of what you measured.
- Materials and Methods => Cryo-EM Sample preparation | Include the blotting conditions for the vitrobot (temperature, humidity, blot force, blot time).
- Materials and Methods => Cryo-EM Sample preparation | were these grids prepared without glow discharge ? If they were treated before, report this.
- Materials and Methods => Data Collection | did the Titan Krios at Southern University of Science and Technology and Shuimu BioSciences have K2 or different detector ?
- Materials and Methods => Data Processing | how many ab initio and heterogeneous classes were used ?
- Materials and Methods => Data Processing | was B-factor sharpening ever applied before model building ?

- Figure 2C,D&E can you show how the map fits with the atomic model here ? Do the Q-scores support where these side chains are ?
- Figure 4C => on this native PAGE was a native marker or another standard used ? If so please make it visible.
- Figure 4D => for the gel of the glycerol gradient, how was this detected ? Coomassie also or something sensitive (e.g. silver stain / SYPRO ruby).
- Figure 5D => on this native PAGE was a native marker or another standard used ? If so please make it visible.
- Figure 5E => it appears that the top (maxima ~6.7 ml) is cut off. Can this be improved by increasing the Y-axis ?

- Figure S1E&F => should be some methods on how this negative stain was carried out.
- Figure S3 = is "1,000 k particles" 1 million ? If so it can expressed as "~1 m".
- Figures S2, S3 => your methods mention "Heterogeneous refinement was performed to pick the best 3D class, followed by homogeneous refinement, non-uniform refinement (Punjani et al, 2020) and local refinement to improve the resolution" this is not reflected in either workflow. What happened to the number of particles, GS-FSC etc ? The differences between S2, S3, and S4 should be reflected in the methods. Also S4 is the only one with a refinement mask.

Point-by-Point Response

Reviewers' comments:

The referees appreciate the importance of the subject and the potential interest of your new findings. However, they all raise a number of significant concerns that would need to be addressed before publication would be warranted. These issues mostly concern missing method details of the structural and biophysical analyses, and insufficient use of cryo-EM processing approaches. There are also concerns about mutation verification and additional/better mutations being needed to validate structures. Finally, the referees mandate better presentations both in the text and the figures.

We thank the reviewers and editor for kind comments and our point-by-point response is the following:

Referee #1:

1. Clarity and Structure: The manuscript is notably difficult to follow, owing to the authors' brevity in explaining complex protein complexes. To enhance readability, the manuscript requires streamlining, and the individual complexes warrant better rationalization.

Response:

1) We are regretful that the brevity in explaining the structure of unneddylated CRL2^{FEM1B} complexes reduced readability. Thus, we have rephrased the paragraph in **lines 101-106** as

suggested. And our manuscript now is following a streamlining from structure of unneddylated CRL2^{FEM1B} (inactive) to that of neddylated CRL2^{FEM1B} (active).

“Using conventional single particle analysis cryo-EM method, we solved the structure of purified recombinant CRL2^{FEM1B} to 4.08 Å resolution (SI Appendix, Fig. S2A and B). The asymmetric super-complex is shown in Fig. 1A and B. The asymmetric super-complex consists of three protomers, but only protomer 1 and protomer 2 were well resolved. Protomer 3 was partially obscured and requires a lower contour threshold level to be visible. (Fig. 1B).”

2. The use of cartoon representations could replace lengthy, confusing nomenclature and make the content more accessible to readers not deeply versed in the CRL field.

Response:

1) We have described these structures with cartoon representations which exhibited the activation and substrate recognition process of CRL2^{FEM1B}, as shown in Fig.8F.

Response Fig.1. (F) Model of the activation and substrate recognition process of CRL2^{FEM1B}.

Each protomer is colored separately in cyan, yellow and red. RBX1, NEDD8, Ub, E2 and substrates are colored in green, purple, pink, magenta, brown and orange, respectively.

3. For instance, the second paragraph of the introduction abruptly introduces numerous complexes and facts, making it challenging for readers to understand the logical connections to the study's core focus in the third paragraph.

Response:

1) In the introduction, we removed complexes and facts that are less related to our topic and reorganize the logical connections in **lines 41-51** as suggested.

“For example, studies of neddylated CRL1^{FBXW7} and ARIH1 hetero-dimeric E3-E3 super-assemblies revealed how two types of E3s co-evolved to transfer ubiquitin to various substrates with folded structures or of limited lengths (Horn-Ghetko et al, 2021). It has also been reported that CUL2 assembles hetero-dimeric CRLs with CUL4A (which is involved in the progression of Alzheimer's disease) (Yasukawa et al, 2020), in addition to forming a homo-dimeric CRL2^{VHL} complex in vivo (Chung et al, 2006; Merlet et al, 2009). Although homo-dimeric Von Hippel Lindau protein (pVHL) could not be detected in vitro, these studies provide strong evidence that oligomerization of CRLs occurs not only via substrate recognition proteins but also via adaptors, Cullin scaffolds, or their combinations (Bulatov & Ciulli, 2015). However, the physiological role of oligomerization of CRL2s is not yet well understood.”

4. However, this comment refers not only to these exemplified sections but the entire manuscript which lacks description of results as well as rational for individual approaches.

Response:

1) We appreciate the reviewer's suggestion. In the revision, we have provided more descriptions of results and reasons for performing individual experiments, such as **in lines 163-167, 170-173, 236-259**. Take the following paragraph as an example:

“To investigate if dimerization of N8-CRL2^{FEM1B} is critical for ubiquitylation of BEX2, we again performed experiments on the oligomerization deficient mutants FEM1B^{F549R}, FEM1B^{F549S}, FEM1B^{F549T}, FEM1B^{Y275R/L278R} and FEM1B^{Δ546-553} (FEM1B^{DEL}), as N8-CRL2^{FEM1B}-BEX2 complex adopted the similar conformation as protomer 1 and protomer 2 of CRL2^{FEM1B}. Consistent with our expectation, MP experiment results indicated that CRL2^{FEM1B} oligomerization deficient mutants mainly formed monomers in the solution except FEM1B^{F549T} (Table 3, SI Appendix, Fig. S8H-L). We then performed *in vitro* ubiquitylation assays at room temperature for 20 min and assessed the ubiquitylation level of MBP-BEX2 and FNIP1/FLCN with N8-CRL2^{FEM1B} and neddylated mutants by western blotting using MBP and FNIP1 antibodies. The *in vitro* ubiquitylation experiments revealed that oligomerization deficient mutants except FEM1B^{F549T} had significantly lower ubiquitylation activities toward BEX2 than wild-type N8-CRL2^{FEM1B}, whereas these mutants did not affect the ubiquitylation of FNIP1/FLCN significantly (Fig. 7A-C). We also assessed the time-resolved ubiquitylation level of MBP-BEX2 and FNIP1/FLCN with N8-CRL2^{FEM1B}, N8-CRL2^{FEM1BDEL} and CRL2^{FEM1B}, respectively (Fig. 7D-G). The plots from the chromogenic western blotting analysis of the ubiquitylated products of MBP-BEX2 confirmed that N8-CRL2^{FEM1B} had higher enzyme activity than N8-CRL2^{FEM1BDEL} and CRL2^{FEM1B} toward MBP-BEX2 (Fig. 7H). As expected, the plots from the chromogenic western blotting analysis of the ubiquitylated products of FNIP1/FLCN indicated that the oligomerization deficient mutant N8-CRL2^{FEM1BDEL} had higher enzyme activity than N8-CRL2^{FEM1B} and CRL2^{FEM1B} (Fig. 7I). Then we performed grating-coupled interferometry experiments on oligomerization deficient mutants and wild-type CRL2^{FEM1B}. The results suggested that mutations didn't influence the process of substrate recognition (Table 2, SI Appendix, Fig. S9)”

5. Quality of Structures: The resolved structures, as presented, are of low quality-appearing smeary and poorly resolved. This is likely due to the simplistic data processing methods and the insufficient number of particles employed. Advanced computational tools like 3DFlex or DeepEMhancer, or a more meticulous 3D classification, could enhance the analysis. While the flexibility of CRL complexes is acknowledged, their oligomeric states appear more stable and deserve better analytical rigor. Also the models are not well done. Many side chains are actually out of density while the density would be available. This is especially critical in the interfaces. If amino acids should be interpreted, as the authors do, the fits have to be reasonable. I am also lacking the customary "Table 1" detailing the acquisition, image processing and modelling including its quality control. This can not be missed in any cryoEM paper and should be included.

Response:

- 1) We thank the reviewer's suggestion. For cryo-EM data processing, we have used TOPAZ for deep-learning particle picking and DeepEMhancer for postprocessing as shown in revised SI Appendix Fig. S2, Fig. S3 and Fig.S4. As for 3DFlex reconstruction and 3DVA, unfortunately, they were not actually helpful in our cases.
- 2) In most cases, oligomerization could stabilize the CRL complexes. For example, the FEM1B subunit in protomer 1 of CRL2^{FEM1B} complex is more stable than FEM1B subunit in N8-CRL2^{FEM1B}-BEX2 and N8-CRL2^{FEM1B}-FNIP1/FLCN complex. Because FEM1B subunit of protomer 1 is suppressed by other protomer in CRL2^{FEM1B} complex. However, as the subunits of CRL have noninterference motions in many directions, the effect of stabilization by oligomerization largely depends on the choices of interfaces. As such, we could find evidences like DOI: <https://doi.org/10.15252/emj.2021108008> and DOI:

<https://doi.org/10.1038/s41467-019-11772-y> to prove that sometimes it's still challenging to resolve high-resolution cryo-EM structures of CRL complexes.

3) 'Table 1' is now provided as required.

Supplementary Table 1

Cryo-EM data collection, refinement and validation statistics

	CRL2 ^{FEM1B}	N8-CRL2 ^{FEM1B} ₁	N8-CRL2 ^{FEM1B} ₂
PDB ID	8IJ1	BEX2	FNIP1
EMDB ID	EMD-35461	8JE1	8JE2
	EMD-35461	EMD-36182	EMD-36183
Data collection and processing			
Magnification	130,000 ×	96,000 ×	105,000 ×
Voltage (kV)	300	300	300
Electron exposure (e ⁻ /Å ²)	50	49.9	50
Defocus range (μm)	-1.5 to -2.0	-1.5 to -2.0	-1.5 to -2.0
Pixel size (Å)	0.92	0.86	0.83
Symmetry imposed	C1	C1	C1
Initial particle images (no.)	146,719	1,049,008	2,358,072
Final particle images (no.)	75,653	210,217	333,541
Map resolution (Å)	4.20	3.95	3.63
FSC threshold	0.143	0.143	0.143
Map resolution range (Å)	56.54-3.43	50.35-1.82	60.01-1.75
Refinement			
Initial model used (PDB code)	5N4W,6LBF	5N4W,6LBF	5N4W,7ROY
Model resolution (Å)	4.3	4.2	4.2
FSC threshold	0.5	0.5	0.5
Model resolution range (Å)	250-4.1	250-2.4	250-2.0
Map sharpening B factor (Å ²)	-56.74	-142.80	-132.85
Model composition			
Non-hydrogen atoms	25220	17508	9912
Protein residues	3172	2165	1237
Ligands	4	1	1
B factors (Å²)			
Protein	157.75	138.62	109.07
Ligand	216.86	139.48	65.89
R.m.s. deviations			
Bond lengths (Å)	0.003	0.003	0.003
Bond angles (°)	0.596	0.715	0.666
Validation			
MolProbity score	1.74	1.97	1.87
Clashscore	10.73	11.74	12.62
Poor rotamers (%)	0.04	0.00	0.09
Ramachandran plot			
Favored (%)	96.88	94.32	96.23
Allowed (%)	3.09	5.68	3.77
Disallowed (%)	0.03	0	0

Response Fig.2. Supplementary Table 1 for cryo-EM data collection, refinement and validation statistics.

6. Mutation Verification: The mutations employed for interface validation seem drastic. For instance, changing Leucine or Phenylalanine to Arginine results in a significant shift from hydrophobic to charged properties. A subtler mutation like Leucine to Serine or Threonine could offer a more convincing argument. There is also a lack of quality control to confirm that these mutations don't alter the overall protein fold. The inclusion of at least one stability assay is essential for demonstrating that the mutations specifically disrupt the binding interface. Quantitative analyses, such as affinity measurements, are also desirable.

Response:

1) To address this question, we performed *in vitro* ubiquitylation assay of FNIP1/FLCN with wild-type and mutant FEM1B variants, such as F549R, F549S, F549T, DEL and Y275R/L278R. Since FNIP1/FLCN is ubiquitylated by monomeric N8-CRL2^{FEM1B}, efficient ubiquitylation of FNIP1/FLCN by FEM1B mutants could indicate that these mutations do not impair their E3 ligase activities and therefore do not alter the overall protein fold as shown in Fig. 7C. However, as shown in Fig. 7C, Y275R/L278R showed lower enzyme activity towards FNIP1/FLCN. Therefore, we believe that mutations on Y275/L278 might have negative impact on the ubiquitylation process.

Response Fig.3. (C) The results of *in vitro* ubiquitylation of FNIP1/FLCN by wild-type N8-CRL2^{FEM1B} and its oligomerization deficient mutants suggest mutants F549R, F549T, F549S and DEL have no negative impact on the enzyme activity while Y275R/L278R shows lower ubiquitylation efficiency towards FNIP1/FLCN.

2) We measured the binding affinity of MBP-BEX2 and FNIP1/FLCN towards wild-type and mutant CRL2^{FEM1B} complexes as shown in Table 2 and SI Appendix Fig. S9. The experimental results suggest that these mutations have no negative effect on substrate binding.

Table 2

The effect of mutations of CRL2FEM1B on binding affinities of substrates

Substrate	Sample	Binding Affinity
FNIP1	CRL2FEM1B-WT	46.254 nM
	CRL2FEM1B-DEL	4.719 nM
	CRL2FEM1B-F549R	8.987 nM
	CRL2FEM1B-F549S	17.485 nM
	CRL2FEM1B-F549T	33.279 nM
	CRL2FEM1B-Y275R/L278R	3.126 nM
BEX2	CRL2FEM1B-WT	44.666 nM
	CRL2FEM1B-DEL	6.624 nM
	CRL2FEM1B-F549R	14.024 nM
	CRL2FEM1B-F549S	22.963 nM
	CRL2FEM1B-F549T	109.396 nM
	CRL2FEM1B-Y275R/L278R	3.990 nM

Response Fig.4. Table 2. Binding affinities of substrates to wild-type and mutants of CRL2^{FEM1B} measured by grating-coupled interferometry.

3) We performed thermal stability assays with wild-type and mutant CRL2^{FEM1B} complexes as shown in Table1 and SI Appendix Fig. S11 (n=3). The results showed that these mutations did not influence the thermal stability of either CRL2^{FEM1B} or FEM1B-ELOB-ELOC.

Table 1
The effect of mutations on CRL2FEM1B and MBP-FCB protein stability (n=3)

Samples	T _m (°C)		
CRL2FEM1B-WT	40.89±0.2	56.28±0.07	
CRL2FEM1B-DEL	47.57±0.25	55.21±0.05	61.27±0.08
CRL2FEM1B-F549R	50.51±0.03	59.27±2.54	
CRL2FEM1B-F549S	50.4±0.13	60.96±0.62	
CRL2FEM1B-F549T	53.78±0.03		
CRL2FEM1B-Y275R/L278R	48.91±0.05		
MBP-FCB-WT	46.78±0.01		
MBP-FCB-DEL	47.23±0.05		
MBP-FCB-F549R	46.58±0.02		
MBP-FCB-F549S	46.69±0.02		
MBP-FCB-F549T	46.7±0.02		
MBP-FCB-Y275R/L278R	47.69±0.02		

Response Fig.5. Table 1. Protein stability of CRL2^{FEM1B} and MBP-FEM1B-ELOC-ELOB measured by NanoDSF.

7. Oligomeric State Assessment: The mass photometry data reveal only minimal amounts of oligomers, raising questions about their actual functional significance. Especially when going to assays.

Response:

1) We thank the reviewer for this great question. Here we think that the oligomerization of CRL2^{FEM1B} is a concentration dependent process. As the mass photometry experiments required low concentration of target proteins, we performed the mass photometry experiments with CRL2^{FEM1B} at a concentration of 10 nM which was 100 times less than the 1 μM concentration of E3s in the ubiquitylation assays. This might be the reason why we could detect strong bands of dimeric and trimeric CRL2^{FEM1B} in the blue-native page rather than monomeric CRL2^{FEM1B}. So we think that results of the mass photometry experiments

mainly show us the oligomerization abilities and molecular weights of CRL2^{FEM1B} but do not reflect the actual ratio of different oligomers during ubiquitylation.

8. Also, the native page does not provide clearer evidence. Alternative methods like SEC-MALS might provide a more robust evaluation of oligomeric states in these complexes. Additionally, the mass photometry seems to have issues judging by the negative peaks in the assay. When using well prepared sample and surface these artifacts should not occur.

Response:

- 1) The results of mass photometry and negative staining microscopy and our experiences suggest that CRL2^{FEM1B} samples are heterogenous with unavoidable degradations and contaminations which could be removed by GraFix. Meanwhile, because N8-CRL2^{FEM1B} was produced by *in vitro* neddylation assay, there are contaminations such as UBE2M, UBA3/NAE1. We found out that SEC-MALS could not do a good job in separating oligomeric CRL2^{FEM1B} and we could only get an average MW which was not accurate. As for N8-CRL2^{FEM1B}-FNIP1/FLCN complex, we believed that it was suitable for SEC-MALS because FNIP1/FLCN induced homogenous N8-CRL2^{FEM1B}-FNIP/FLCN complex. Meanwhile, SEC-MALS needs a relative higher concentration of pure proteins which is hard to achieve for neddylated CRLs.
- 2) Therefore, we again used freshly-prepared samples to perform mass photometry measurements as shown in Table 3 and SI Appendix Fig. S8 (n=3). Unfortunately, we noticed that negative peaks are noises caused by our HBS buffer, which could hardly be removed by filtering twice through 0.2 μm filters. Because there was no such peak when we measured other samples with PBS buffer. In neddylated CRL2^{FEM1B} related samples, low MW peaks

are contaminations like UBE2M and UBA3/NAE1. In CRL2^{FEM1B} related samples, low MW peaks are mostly unavoidable degradations.

Table 3
Molecular weights of wild-type and mutants of (N8-)CRL2FEM1B
and E3-substrate complexes measured by mass photometry experiments

Sample	Molecular Weight (kDa)		
CRL2FEM1B-WT	191±1	414±31	576±3
CRL2FEM1B-DEL	191±9		
CRL2FEM1B-F549R	189±3	386±13	
CRL2FEM1B-F549T	186±0	384±21	
CRL2FEM1B-F549S	187±3	375±11	
CRL2FEM1B-Y275RL278R	188±11		
N8-CRL2FEM1B-WT	197±5	402±13	
N8-CRL2FEM1B-DEL	197±27		
N8-CRL2FEM1B-F549R	203±28		
N8-CRL2FEM1B-F549T	190±18	395±11	
N8-CRL2FEM1B-F549S	192±17		
N8-CRL2FEM1B-Y275RL278R	183±7		
N8-CRL2FEM1B-WT-BEX2	254±79	517±36	
N8-CRL2FEM1B-WT-FNIP1/FLCN	196±31	380±12	

Response Fig.6. Table 3. Molecular weights of wild-type and mutants of CRL2^{FEM1B} and N8-CRL2^{FEM1B} and BEX2 or FNIP1/FLCN binding to N8-CRL2^{FEM1B} measured by mass photometry experiments.

3) However, as three times of mass photometry measurements gave us consistent results of molecular weights of mentioned complexes, we believe that they are accurate enough to support our conclusion.

9. Ubiquitylation Assay Description: The ubiquitylation assay is pivotal to the study but is poorly described. Additional details, especially relating to critical figures like Figure 5B, are needed. Visual aids illustrating the constructs used could improve understanding.

Response:

1) We have provided cartoon representations of constructs of **Figure 5B** as suggested.

Response Fig.7 The construct of MBP-FNIP1^{degron} is shown in cartoon representations. CUL2-ELOB-ELOC-FEM1B, RBX1, NEDD8, Ub, E2 and FNIP1 are colored in cyan, green, purple, pink, magenta and orange, respectively.

2) We have provided more details about the ubiquitylation assay as suggested in **line 240-256**.

“Consistent with our expectation, MP experiment results indicated that CRL2^{FEM1B} oligomerization deficient mutants mainly formed monomers in the solution except FEM1B^{F549T} (Table 3, SI Appendix, Fig. S8H-L). We then performed *in vitro* ubiquitylation assays at room temperature for 20 min and assessed the ubiquitylation level of MBP-BEX2 and FNIP1/FLCN with N8-CRL2^{FEM1B} and neddylated mutants by western blotting using MBP and FNIP1 antibodies. The *in vitro* ubiquitylation experiments revealed that oligomerization deficient mutants except FEM1B^{F549T} had significantly lower ubiquitylation activities toward BEX2 than wild-type N8-CRL2^{FEM1B}, whereas these mutants did not affect the ubiquitylation of FNIP1/FLCN significantly (Fig. 7A-C). We also assessed the time-resolved ubiquitylation level of MBP-BEX2 and FNIP1/FLCN with N8-CRL2^{FEM1B}, N8-CRL2^{FEM1BDEL} and CRL2^{FEM1B}, respectively (Fig. 7D-G). The plots from the chromogenic western blotting analysis of the ubiquitylated products of MBP-BEX2 confirmed that N8-CRL2^{FEM1B} had higher enzyme activity than N8-CRL2^{FEM1BDEL} and CRL2^{FEM1B} toward MBP-BEX2 (Fig. 7H). As expected, the plots

from the chromogenic western blotting analysis of the ubiquitylated products of FNIP1/FLCN indicated that the oligomerization deficient mutant N8-CRL2^{FEM1BDEL} had higher enzyme activity than N8-CRL2^{FEM1B} and CRL2^{FEM1B} (Fig. 7I).”

10. Again, it is vital to be quantitative and perform assays that can give physical parameters rather than just an endpoint visualization.

Response:

1) To address this question, we repeated the ubiquitylation assay and quantified the ubiquitylation levels of MBP-BEX2 and FNIP1/FLCN using ImageJ as shown in **Figure 7H** and **I**.

Response Fig.8. (H, I) Plots of the fraction of substrates that have not been converted to ubiquitylated products against time for ubiquitylation reactions containing CRL2^{FEM1B}, N8-CRL2^{FEM1B} or N8-CRL2^{FEM1BDEL} (as shown in D-G). The plots of ubiquitylated substrates were quantified using ImageJ. The data were fit to one-phase exponential decay function with time constant parameter ($y=A1*\exp(-x/t1)+y0$) to obtain rates of ubiquitylation.

11. The manuscript refers to Figures S3 and S5 when discussing the cryo-EM structure of the N8-CRL2FEM1B-BEX2 complex. These figures appear to detail the workflow rather than the actual structure, warranting clarification from the authors.

Response:

- 1) We have provided a main figure (**Figure 6A**) showing the cryo-EM map of N8-CRL2^{FEM1B}-BEX2 colored by subunits to address this problem.

Response Fig.9. (A) Cryo-EM densities of N8-CRL2^{FEM1B}-BEX2 complex with each of its subunits assigned a different color.

12. Methodological Details: The Grafix procedure described in the methods section lacks critical details such as the type of gradient, rotor, and GA concentration used, which could affect replicability and validation of the results.

Response:

1) We updated the information about the Grafix procedure in the section of Cryo-EM Sample Preparation in **lines 552-555**.

“For GraFix, we followed the same protocols as for glycerol gradient ultracentrifugation, except that we added 0.15% (v/v) GA to the bottom buffer. The bottom buffer consisted of 50 mM HEPES (pH 7.5), 50 μ M ZnCl₂, 100 mM NaCl, 30% (v/v) glycerol and 0.15% (v/v) GA.”

13. The term "E1 ubiquitin ligase" is incorrectly used in the first paragraph; E1 refers to the ubiquitin-activating enzyme and not a ligase.

Response:

1) We corrected it as “ubiquitin activating enzyme (E1)” in **line 28**.

14. The current citation style hinders readability by placing all citations at the end of sentences. Placing each citation immediately after the relevant claim would improve clarity.

Response:

1) We have revised the citations in **lines 19-24** and **53-55** to improve clarity. Other citations that we placed at the end of sentences are mainly claiming same points, respectively.

“CRLs have crucial roles in controlling the cell cycle (Jang et al, 2020), hypoxia signaling (Maxwell et al, 1999), reactive oxygen species clearance (Kensler et al, 2007) and DNA repair

(Ferretti et al, 2016), all of which are pivotal processes regulating several disorders including cancer (Zhao & Sun, 2013), Alzheimer’s disease (Potjewyd & Axtman, 2021), and delayed development (Jiang et al, 2012), as well as tissue response to ionizing radiation (Fouad et al, 2019).”

“FEM1B is the substrate recognition protein of the oligomeric CRL2^{FEM1B} E3 ubiquitin ligase (Dankert et al, 2017; Koren et al, 2018; Lin et al, 2018; Wang et al, 2016) that has been implicated in sex determination (Starostina et al, 2007), apoptosis (Chan et al, 2000), colon cancer (Subauste et al, 2010) and Alzheimer’s disease (Crist et al, 2021).”

15. Grammatical issues, including missing articles, should be rectified for a polished, professional manuscript.

Response:

1) We apologize for those grammatical issues and we have gone through and revised the manuscript as suggested.

Referee #2:

1. Mass photometry was used to assess oligomerization of Cul2Fem1B and mutants. Additional data from multi-angle light scattering, ultracentrifugation and direct protein visualization of bands on native gels is reported for select samples. Unfortunately, these above techniques were not uniformly applied to all complexes which makes it difficult to assess the validity of their conclusions.

Response:

1) To address this question, we performed mass photometry measurements with neddylated/unneddylated CRL2^{FEM1B} and mutants as shown in Table 3 and SI Appendix Fig.S8 (n=3). As shown in Response Fig.6.

2. For example, Fem1B interface mutants F549R, Y277R/L278R and a loop deletion (DEL1 Δ 546-553) disrupt the trimeric state completely and dimeric form partially? (DEL1 mutant only) in the mass photometry assay (Fig Supp.8). Functionally, only the DEL1 and F549R mutant were tested and shown to retain FNIP1 binding (Fig.S9).

Response:

1) To address this problem, we measured the binding affinity of wild-type CRL2^{FEM1B} and its mutants F549R, F549S, F549T, Y275R/L278R and DEL to BEX2 and FNIP1/FLCN as shown in Table 2 and SI Appendix S9. As shown in Response Fig.4.

3. These mutants were only tested for ubiquitination of the MBP-BEX substrate (Fig 6G) but not the FNIP1/FLCN substrate. Is the monomeric DEL1 mutant still functional to ubiquitinate FNIP1/FLCN?

Response:

1) To answer this question, we performed ubiquitylation assays of CRL2^{FEM1B} and its mutants mentioned above against MBP-BEX2 and FNIP1/FLCN as shown in Fig.7A-C.

Response Fig.10. (A, B) The results of *in vitro* ubiquitylation of BEX2 by wild-type N8-CRL2^{FEM1B} and its oligomerization deficient mutants suggest that dimerization of N8-CRL2^{FEM1B} is necessary for ubiquitylation of BEX2. Samples were analyzed by western blotting using antibodies against MBP, FEM1B, CUL2, UBA1, UBE2D3 or UBE2R1, respectively.

(C) The results of *in vitro* ubiquitylation of FNIP1/FLCN by wild-type N8-CRL2^{FEM1B} and its oligomerization deficient mutants suggest mutants F549R, F549T, F549S and DEL have no negative impact on the enzyme activity while Y275R/L278R shows lower ubiquitylation efficiency towards FNIP1/FLCN.

4. The role of Cul2-Neddylated on the oligomeric state was also examined. A neddylated-Cul2Fem1B complex was able to ubiquitinate each substrate examined (although the unneddylated version was not tested). No trimeric version of the N8-Cul2FEM1B complex was observed by mass photometry.

Response:

1) We also performed *in vitro* ubiquitylation assays of unneddylated CRL2^{FEM1B} complex against BEX2 and FNIP1/FLCN as shown in Fig. 7E, G, H and I. The results show that unneddylated CRL2^{FEM1B} have minor enzyme activities comparing to neddylated CRL2^{FEM1B}

and its mutants. It suggests that dimerization and neddylation are both required for ubiquitylation of MBP-BEX2.

Response Fig.11. (E, G) Time-resolved *in vitro* ubiquitylation assays of MBP-BEX2 and FNIP1 by CRL2^{FEM1B} show that CRL2^{FEM1B} has minor enzyme activity towards either MBP-BEX2 or FNIP1.

(H, I) Plots of the fraction of substrates that have not been converted to ubiquitylated products against time for ubiquitylation reactions containing CRL2^{FEM1B}, N8-CRL2^{FEM1B} or N8-CRL2^{FEM1BDEL}. The plots of ubiquitylated substrates were quantified using ImageJ. The data were fit to one-phase exponential decay function with time constant parameter ($y=A1*\exp(-x/t1)+y0$) to obtain rates of ubiquitylation.

5. Further evidence of the impact Neddylated Cul2 has on complex formation was examined by a cryo-EM structure of Neddylated Cul2^{FEM1B} in complex with MBP-Bex2. Given the Neddylated site (lys689) is at a Cul2-Cul2 interface of the unneddylated-oligomeric complex (Fig.4a), a description of the structural changes that occur when this site on Cul2 is Neddylated was lacking.

Response:

1) To answer this question, we revisited the density maps of N8-CRL2^{FEM1B}-BEX2 complex and unneddylated CRL2^{FEM1B} complex. We found that there was an extra density connected to the WHB domain of protomer 2 in the N8-CRL2^{FEM1B}-BEX2 complex, which did not exist in the cryo-EM map of the unneddylated CRL2^{FEM1B} complex. We believed that the extra density corresponded to the density of NEDD8. These results suggest that this conformation could exist after neddylation. However, it remains unclear whether neddylation could interrupt the interface around Lys689. In this case, we believe that neddylation may induce the dissociation of protomer 3 in the trimeric CRL2^{FEM1B} complex. Thus, we have corrected **lines 179-181** and **Figure 4A** in the section ‘Neddylation alters the oligomeric state of CRL2^{FEM1B} E3 Ubiquitin Ligase’.

“We hypothesized that CRL2^{FEM1B} neddylation might impair the protomer 3-protomer 2 interaction, as protomer 3 bound protomer 2 weakly in a ‘head-to-tail’ orientation through the WHB domain and RBX1 (Fig. 4A).”

Response Fig.12. (A) Protomers 2 and 3 form contacts in a ‘head’ to ‘tail’ conformation in which the substrate recognition pocket of one protomer is occupied by the RBX1 and flexible WHB domain of CUL2 of the other protomer. Broken black circled regions correspond to RBX1 subunit and WHB domain of protomer 2 and protomer 3, respectively.

6. Finally, a structure was determined in the presence of FNIP1/FLCN. The extent of map contouring to support substrate engagement was not convincing to this reviewer. Perhaps better datasets will provide more evidence to support a full Cul2FEM1b-FNIP1/FLCN complex structure. The authors state that monomeric Cul2FEM1B is capable of ubiquitinating this intact protein substrate yet there is no functional data demonstrating this with a monomeric-interface mutant such as the DEL1.

Response:

1) To answer this question, we performed *in vitro* ubiquitylation assay of all mutants of N8-CRL2^{FEM1B} against FNIP1/FLCN as shown in **Figure 7C**. And in fact, N8-CRL2^{FEM1BDEL} had higher enzyme activity towards FNIP1/FLCN than N8-CRL2^{FEM1B} as shown by **Figure 7E**. **As shown in Response Fig.10C, Response Fig.11I and Response Fig.13.**

Response Fig.13. Time-resolved *in vitro* ubiquitylation assays of FNIP1 by N8-CRL2^{FEM1B} and N8-CRL2^{FEM1BDEL} show that N8-CRL2^{FEM1BDEL} is more active to FNIP1 comparing to N8-CRL2^{FEM1B}.

2) Meanwhile, we revisited the cryo-EM data with new algorithms such as TOPAZ picking and deepEMhancer. However, we could only obtain minor improvements in the resolution of Cryo-EM map. We think that it is mainly caused by the flexibility of FNIP1/FLCN as it is composed of many loops predicted by Alphafold2.

7. How can uniform Neddylaton on K687 of Cul2 be consistent with dimerization given the view provided in Figure 4? This structural hypothesis appears inconsistent with complete Cul2-Neddylaton demonstrated in Fig 4B yet a prominent dimeric band on a native gel in Fig 4C.

Response:

1) After revisiting the cryo-EM maps of CRL2^{FEM1B} and N8-CRL2^{FEM1B}-BEX2 complex, we believe that K689 is sufficiently accessible for neddylaton and there is not enough evidence to discuss whether neddylaton on K689 could disrupt this interface.

8. Authors claim that dimeric Cul2FEM1B is active for ubiquitination of MBP-Bex and MBP-FNIP1-degron substrates while Cul2FEM1B monomer ubiquitinates FNIP1/FLCN. Oligomeric states of these Cul2FEM1B-substrate complexes are compared by different methods however (Blue native PAGE versus SEC-MALS and mass photometry respectively). Can the authors demonstrate the dimeric nature of the BEX or FNIP1-degron complex by SEC-MALS and or mass photometry for a direct comparison? Does a monomeric Cul2FEM1B complex ubiquitinate FNIP1/FLCN since this was only attempted for the MBP-Bex substrate.

Response:

- 1) To answer these questions, we performed mass photometry experiments on all CRL2^{FEM1B}-substrate complexes as shown in Table 3 and SI Appendix Fig.S8 M and N (n=3). We also performed *in vitro* ubiquitylation assay on oligomerization-deficient mutants of CRL2^{FEM1B} against FNIP1/FLCN as shown in Figure 7C. As shown in Response Fig.6 and Response Fig.10C.
- 2) Together with Figure 7A-1, these results suggest that N8-CRL2^{FEM1B} has higher enzyme activity towards MBP-BEX2 than N8-CRL2^{FEM1BDEL} while N8-CRL2^{FEM1BDEL} is more active to FNIP1/FLCN comparing to N8-CRL2^{FEM1B}. As shown in Response Fig.11.

9. EM map not clearly visible in Figure 6E.

Response:

- 1) We have adjusted the transparency level of the EM map to make it more visible.

Response Fig.14. (F) Atomic models fitted in cryo-EM map shows details of the hydrophobic interactions between the N-terminal α -helix of FEM1B (residues 269-284) in protomer 1 and the C-terminal ankyrin repeats of FEM1B (residues 546-588) in protomer 2.

10.Oligomeric interface mutants investigated: Y277R/L278R - only L278 is visible in the interface shown in Figure 3B/Fig S7B. Why was Y277 targeted for mutagenesis since this sidechain does not appear oriented towards protamer 2 (while Y275 appears a better candidate on this basis). Were any other interface mutants tested and not shown in the manuscript?

Response:

- 1) Thank you very much for reminding us. The mutation of Y277 was misnamed and in fact Y275 along with L278 was mutated to Y275R and L278R. However, as shown in **Fig.7C**, Y275R/L278R had lower enzyme activity towards FNIP1/FLCN. We believe that mutations at Y275/L278 might have negative impact on the ubiquitylation process. As shown in **Response Fig.10C**.

2) Honestly, there were no other interface mutants tested. We chose Y275, L278 and F549 mainly based on their high distributions to the hydrophobic interface and low evolutionary conservation which were typical and possibly had minor influence to the protein stability.

11. Any of the cross links from Gradient Fixation visible in the structures? Was any structural classes obtained from non Grafix datasets that are consistent with the oligomeric super-complexes described?

Response:

1) Yes, we have observed densities that might correspond to cross-links in the structures such as K62 and K115 of CUL2 and K598 of FEM1B in the cryo-EM map of N8-CRL2^{FEM1B}-BEX2 complex. However, no densities corresponding to cross-links have been observed in the oligomerization interface. Due to the heterogeneity of oligomeric CRL2^{FEM1B} as shown by negative staining micrographs and mass photometry, we were not confident about resolving the non-Grafix cryo-EM structures of oligomeric CRL2^{FEM1B}. Considering the cryo-EM structures resolved and the results of mass photometry and negative staining microscopy together, we believe that Grafix mainly helped remove the unavoidable degradations and contaminations.

Referee #3:

1. Abstract => be careful using the term "the first" Maniaci 2017 (<https://doi.org/10.1038/s41467-017-00954-1>) have induced oligomerization for CRL2VHL and since then other homo- and

hetero-PROTACs have been reported. Your introduction goes on to cite numerous other examples.

Response:

We have rephrased this sentence in lines 10-11.

2. Introduction => first sentence "Ring" is usually "RING" for E3 ubiquitin ligases (Really Interesting New Gene) with RBX1 and RBX2 being the only known for CRLs.

Response:

We have revised those words in lines 2 and 17.

3. Introduction => "In the proteasomal degradation system or ubiquitin- proteasome system (UPS)" is repetitive.

Response:

We have revised those words in line 25.

4. Introduction => "poly-ubiquitin" can be one word "polyubiquitin" or "polyUb"

Response:

We have revised those words in line 25.

5. Introduction => "E2 ubiquitin ligases" this is usually "E2 ubiquitin-conjugating enzyme (E2) and E3s are the ligases.

Response:

We have revised those words in line 26.

6. Introduction => "E1 ubiquitin ligases (E1s)" usually E1 refers to ubiquitin activating enzyme (E1)

Response:

We have revised those words in line 28.

7. Introduction => "It has also been reported that CUL2 could not only assemble hetero-dimeric CRLs with CUL4A" this wording is confusing. "It has also been reported that CUL2 assembles hetero-dimeric CRLs with CUL4A (which is involved in the progression of Alzheimer's disease) (Yasukawa et al, 2020), in addition to forming homo-dimeric CRL2VHL complex in vivo (Chung et al, 2006; Merlet et al, 2009)."

Response:

Thank you for your suggestion. We have rephrased those parts as suggested in lines 44-47.

8. Introduction => "these studies provide strong evidence that oligomerization of CRLs occurs not only via substrate recognition proteins but also via adaptors, or Cullin scaffolds, or their combinations as well (Bulatov & Ciulli, 2015)." Change to "these studies provide strong evidence that oligomerization of CRLs occurs not only via substrate recognition proteins but also via adaptors, Cullin scaffolds, or their combinations (Bulatov & Ciulli, 2015)."

Response:

Thank you for your suggestion. We have rephrased those parts as suggested in lines 48-50.

9. Results => "by His-Trapping and Q-Column" this could just be "IMAC and anion exchange"
The use of "His-Trapping" is not ideal here.

Response:

Thank you for your suggestion. We have corrected those words as suggested in line 96.

10. Results => "Using standard cryo-EM methods, we solved the structure of purified recombinant trimeric CRL2FEM1B to 7.38 Å resolution" by "standard" do you mean conventional single particle?

Response:

Yes, we mean the conventional single particle analysis cryo-EM method. We have revised this phrase as suggested in line 101.

11. Results => "Cryo-EM structure of CRL2FEM1B dimeric and trimeric super-complex" This section read like a methods section in parts, but regardless should end with a concluding sentence.

Response:

We rephrased that title as "Cryo-EM structures of CRL2^{FEM1B} complex shows asymmetric super-assembly" in line 93.

12. Results => "When the two structures are superimposed, the CUL2-RBX1 scaffolds and adaptor proteins differ slightly, with a root-mean-square deviation of 2.088 Å (SI Appendix, Fig. S6A)" is this RMSD alpha carbons (C α) or something else

Response:

The RMSD is an average measure of how far apart an α -Carbon (C α) atom is in one structure is from the C α atom from the other structure after they are aligned. We are making it clear in the manuscript as suggested in lines 116 and 219.

13. Results => "Furthermore, the loop connecting a-helix D and a-helix E of the CUL2 cullin repeat 1 (which was not visible in previously reported structures (Cardote et al., 2017)), appears to bind the C-terminus of FEM1B through hydrophobic interactions" you can re-word to include "which was not resolved in the previously reported X-ray crystal structure (Cardote et al., 2017)" There seems to have only been one structure reported PDB-5N4W.

Response:

We have rephrased it as suggested in line 140.

14. Discussion => there are great examples of oligomerizing substrate adapters and CRLs. However, given your focus on the C-terminus and RBX1 would it make sense to expand on how E2~Ub or E2~N8 binding to RBX1 in CRL2^{FEM1B} could alter the oligomer ?

Response:

In this manuscript, we mainly focus on the influence of neddylation and substrate recognition on the oligomerization of CRL2^{FEM1B} E3 ubiquitin ligase. Initially, we attempted to resolve the structure of N8-CRL2^{FEM1B} complexed with UBE2D3(C85S)-Ub and FNIP1-FLCN, hoping to provide insights into the substrate ubiquitylation process. However, we could only obtain the stable N8-CRL2^{FEM1B}-FNIP1-FLCN complex, because the binding affinity between UBE2D3(C85S)-Ub complex and N8-CRL2^{FEM1B} was too weak. Therefore, E2~Ub binding to RBX1 in CRL2^{FEM1B} might not significantly affect the oligomerization. Unfortunately, due to the dynamics and low local resolution of RBX1 and C-terminus of CUL2 in our structures, it is difficult to discuss the possible effect of E2~Ub or E2~N8 binding to RBX1.

15. Material and Methods => Experiment model and subject details "E. coli were used for molecular cloning and expression of recombinant FEM1B-ELOB-ELOC and its mutants." Did you use two different strains for E. coli. It is highly important to report the expression strain (e.g. BL-21 Rosetta2).

Response:

In our experiments, we used the DH5 α strain for molecular cloning and the BL21(DE3) strain for expression of those recombinant proteins. We have revised these as suggested in lines 344-345.

16. Materials and Methods => Molecular Cloning and Generation of Constructs "All proteins are of human origin and are encoded by cDNAs amplified from a testis cDNA library. All the proteins and variants referred to were generated using PCR." Were regular BL-21 (DE3) cells used ? If so were there any issues with codons when using human cDNA and an E. coli strain that may lack some of the tRNAs for rare codons ?

Response:

- 1) We used the DH5 α strain of *E. coli* for molecular cloning and generation of constructs.
- 2) We used the BL21(DE3) strain for expression of those recombinant proteins. Fortunately, we did not encounter the problems of rare codons in this case.

17. After reading the Materials and Methods, why is the first section there, it is repeated and expanded on in the following sections.

Response:

In this section, we intended to describe the cell lines used in our experiments.

18. Materials and Methods => Protein purification what scale are we talking about ~1mg, or tens of mg of protein. In general try to also include the size/total column volume of the columns used (e.g. 5 ml His-Trap, 10/300 Superose6, etc).

Response:

We purified proteins from *E. coli* (800 ml LB) and SF9 (500 ml SIM-SF) cells, yielding ~5 mg of MBP-FEM1B-ELOB-ELOC and ~1 mg of CUL2-RBX1 per run. We used the following columns: 10 ml His-Trap, 6 ml Resource Q IEX, 5 ml Strep-trap, Superdex 200 increase 10/300 gl, and Superose 6 increase 10/300 gl. We have added these information to the manuscript as suggested in lines 385-424.

19. Materials and Methods => Protein purification "The CRL2FEM1B ligase complex was eluted using a linear gradient ranging from 150 mM to 300 mM NaCl with buffer Wb (50 mM HEPES pH 7.5, 1 M NaCl and 5 mM DTT)." For how many column volumes was this gradient and at what % did the CRL elute ?

Response:

We eluted at 2 ml/min with a 20 ml linear gradient from 150 mM to 300 mM NaCl. CRL eluted between 190 mM and 280 mM NaCl.

20. Materials and Methods => Western Blotting | can you describe the type of gel used (e.g. 10% SDS-PAGE, 1mm, Bis-Tris, Life Technologies) and also the running buffer (Tris-Glycine, MES, MOPS, etc).

Response:

We used a 4-20% Precast Protein Plus Gel (15 wells, Hepes-Tris) from YEASEN and a Precast Running Buffer for SDS-PAGE. And we have added it to the manuscript as suggested in lines 443-444.

21. Materials and Methods => Mass Photometry | Include the final concentration of what you measured.

Response:

The final concentrations of complexes we measured are 10 nM. We have provided the final concentration as suggested in lines 486.

22. Materials and Methods => Cryo-EM Sample preparation | Include the blotting conditions for the vitrobot (temperature, humidity, blot force, blot time).

Response:

We used the following settings on Vitrobot VI: blot force 1, blot time 1, wait time 15 s, 100% humidity, and 4°C. And we have added this information in lines 502-503.

23. Materials and Methods => Cryo-EM Sample preparation | were these grids prepared without glow discharge ? If they were treated before, report this.

Response:

We glow-discharged the grids with air before use. We have added this information in line 500.

24. Materials and Methods => Data Collection | did the Titan Krios at Southern University of Science and Technology and Shuimu BioSciences have K2 or different detector ?

Response:

The Titan Krios at SUSTech had a K3 detector, while the one at Shuimu BioSciences had a Falcon 4 detector. We have added this information in lines 508-511.

25. Materials and Methods => Data Processing | how many ab initio and heterogeneous classes were used ?

Response:

The cases have different numbers based on the heterogeneity of available particles. For uneddylated CRL2^{FEM1B}, 3 classes were used as shown in SI Appendix Fig. S2. For neddylated CRL2^{FEM1B}-BEX2 complex, a maximum of 6 classes were used as shown in SI Appendix Fig. S3. For neddylated CRL2^{FEM1B}-FNIP1/FLCN complex, 3 classes were used as shown in SI Appendix Fig. S4.

26. Materials and Methods => Data Processing | was B-factor sharpening ever applied before model building ?

Response:

B-factor for sharpening are provided in Supplementary Table 1. And deepEMhancer were used before model building for better interpretation.

Supplementary Table 1

Cryo-EM data collection, refinement and validation statistics

	CRL2 ^{FEM1B}	N8-CRL2 ^{FEM1B} - BEX2	N8-CRL2 ^{FEM1B} - FNIP1
PDB ID	8IJ1	8JE1	8JE2
EMDB ID	EMD-35461	EMD-36182	EMD-36183
Data collection and processing			
Magnification	130,000 ×	96,000 ×	105,000 ×
Voltage (kV)	300	300	300
Electron exposure (e ⁻ /Å ²)	50	49.9	50
Defocus range (µm)	-1.5 to -2.0	-1.5 to -2.0	-1.5 to -2.0
Pixel size (Å)	0.92	0.86	0.83
Symmetry imposed	C1	C1	C1
Initial particle images (no.)	146,719	1,049,008	2,358,072
Final particle images (no.)	75,653	210,217	333,541
Map resolution (Å)	4.20	3.95	3.63
FSC threshold	0.143	0.143	0.143
Map resolution range (Å)	56.54-3.43	50.35-1.82	60.01-1.75
Refinement			
Initial model used (PDB code)	5N4W,6LBF	5N4W,6LBF	5N4W,7ROY
Model resolution (Å)	4.3	4.2	4.2
FSC threshold	0.5	0.5	0.5
Model resolution range (Å)	250-4.1	250-2.4	250-2.0
Map sharpening B factor (Å ²)	-56.74	-142.80	-132.85
Model composition			
Non-hydrogen atoms	25220	17508	9912
Protein residues	3172	2165	1237
Ligands	4	1	1
B factors (Å²)			
Protein	157.75	138.62	109.07
Ligand	216.86	139.48	65.89
R.m.s. deviations			
Bond lengths (Å)	0.003	0.003	0.003
Bond angles (°)	0.596	0.715	0.666
Validation			
MolProbity score	1.74	1.97	1.87
Clashscore	10.73	11.74	12.62
Poor rotamers (%)	0.04	0.00	0.09
Ramachandran plot			
Favored (%)	96.88	94.32	96.23
Allowed (%)	3.09	5.68	3.77
Disallowed (%)	0.03	0	0

Response Fig.15. Map sharpening *B* factors of CRL2^{FEM1B}, N8-CRL2^{FEM1B}-BEX2 and N8-CRL2^{FEM1B}-FNIP1.

27. Figure 2C,D&E can you show how the map fits with the atomic model here ? Do the Q-scores support where these side chains are ?

Response:

Yes, we have provided those figures with atomic models fitted in maps as suggested in **Figure 2C, D and F**. We apologize that we didn't calculate the Q-scores of those side chains. However, we think side chains are properly fitted with visual assessment.

Response Fig. 16. (C) Details of the atomic interactions between the FEM1B BC box and ELOC.

(D) Three-way atomic interactions among the FEM1B Cullin box, ELOC and CUL2.

(F) Details of the atomic interactions between FEM1B and the loop connecting α -helix D and α -helix E of CUL2 Cullin Repeat1 (CR1).

28. Figure 4C => on this native PAGE was a native marker or another standard used ? If so please make it visible.

- Figure 5D => on this native PAGE was a native marker or another standard used ? If so please make it visible.

Response:

We used a NativeMark Unstained Protein Standard from Thermo Fisher Scientific for native PAGEs. Because there are none related samples between our sample and marker, we are providing figures of the gels for review as an alternative.

Fig. 4C

Fig. 5D

Response Fig.17. (4C) Neddylation induces disassociation of trimeric CRL2^{FEM1B} and formation of dimeric N8-CRL2^{FEM1B}. Samples were analyzed by blue native PAGE and stained with Coomassie brilliant blue.

(5D) MBP-BEX2 and MBP-FNIP1^{degron} bind to dimeric N8-CRL2^{FEM1B}. Bands are indicated by dash lines, respectively. Samples were analyzed by blue native PAGE and stained with Coomassie brilliant blue.

29. Figure 4D => for the gel of the glycerol gradient, how was this detected ? Coomassie also or something sensitive (e.g. silver stain / SYPRO ruby).

Response:

It was stained with Coomassie brilliant blue. We have added this information in **lines 857-858**.

30. Figure 5E => it appears that the top (maxima ~6.7 ml) is cut off. Can this be improved by increasing the Y-axis ?

Response:

It has been revised as suggested.

31. Figure S1E&F => should be some methods on how this negative stain was carried out.

Response:

1) The section of negative staining has been added in the methods in lines 466-473.

“Negative staining electron microscopy

The protein samples were prepared by diluting them to 0.03 mg/ml and spinning them at 12000 rpm for 10 minutes at 4 °C. The carbon films were plasma-cleaned for one minute at low setting. Then, 8 µl of protein was pipetted onto the film and left for one minute before blotting with filter paper. The film was rinsed with water and 1% uranium acetate for 10 seconds each and stained with 1% uranium acetate for one minute before blotting again. The negatively stained grids were examined with a JEM-1400PLus electron microscope at 100 kV and an EMSIS CCD camera at 40000 X magnification.”

32. Figure S3 = is "1,000 k particles" 1 million ? If so it can expressed as "~1 m".

Response:

It has been corrected as suggested.

33. Figures S2, S3 => your methods mention "Heterogeneous refinement was performed to pick the best 3D class, followed by homogeneous refinement, non-uniform refinement (Punjani et al, 2020) and local refinement to improve the resolution" this is not reflected in either workflow.

What happened to the number of particles, GS-FSC etc ? The differences between S2, S3, and S4 should be reflected in the methods. Also S4 is the only one with a refinement mask.

Response:

1) We have revised the section of data processing as suggested in lines 528-533 and SI Appendix Fig. S2, S3 and S4 have been revised as well.

“Cryo-EM datasets were processed using cryoSPARC v4.2.1 (Punjani et al, 2017; Rohou & Grigorieff, 2015; Rubinstein & Brubaker, 2015; Stagg et al, 2014; Zheng et al, 2017). Image stacks of unneddylated CRL2^{FEM1B} and neddylated CRL2^{FEM1B}-FNIP1-FLCN complex were aligned using Patch motion correction, and defocus value estimation was performed by Patch CTF estimation. Image stacks of neddylated CRL2^{FEM1B}-BEX2 were aligned using MotionCor2 and defocus value estimation was performed by Patch CTF estimation. Manually selected particles were used to create templates for template-based picking. After several iterations of 2D Classification, particles from the highest-resolution classes were used for 3D ab initio reconstruction. Heterogeneous refinement was performed to pick the best 3D class, followed by homogeneous refinement and non-uniform refinement (Punjani et al, 2020) to improve the resolution. Best particles were then used as template for TOPAZ particle picking (Bepler et al, 2019). After multiple rounds of heterogeneous refinement, best particles were used for homogeneous refinement and non-uniform refinement to improve the resolution.”

Prof. Yuxin Yin
Peking University Health Science Center
Institute of Systems Biomedicine
38 Xue Yuan Road
Beijing, Beijing 100191
China

23rd Jan 2024

Re: EMBOJ-2023-115372R
Structural Insights into the Ubiquitylation Strategy of the Oligomeric CRL2FEM1B E3 Ubiquitin Ligase

Dear Dr. Yin,

Thank you for submitting your revised manuscript to The EMBO Journal. It has now been seen once more by two of the original referees, and I am happy to say that both were generally satisfied with your revisions and responses to the initial comments. However, they still retain a number of presentational issues (see comments below, as well as attached screenshots from referee 3), which I would invite you to incorporate during a final round of minor revision.

In addition, please carefully address also the following editorial issues in the final version:

- Please make sure to reference Appendix Table S1 (cryo-EM data collection) at least once in the main text.
- Please double-check all citations in the reference list, as some of them appear to be still incomplete (lacking page/locator numbers).
- Please rename the Conflict of Interest section into "Disclosure and Competing Interests Statement", in accordance with our updated Guide to Authors (<https://www.embopress.org/competing-interests>)
- As we are switching from a free-text author contribution statement towards a more formal statement based on Contributor Role Taxonomy (CRediT) terms, please remove the present Author Contribution section and instead specify each author's contribution(s) directly in the Author Information page of our submission system during upload of the final manuscript. See <https://casrai.org/credit/> for more information.
- Please rename the "Data and materials availability" section into simply "Data Availability", and delete its first sentence. Please also remove the included reviewer access information at this point, and ensure that data become publicly accessible upon acceptance.
- Please double-check to make sure to all relevant funding information in the manuscript is congruent with the info entered into our submission system. Currently missing in the submission system are : The Shenzhen High-level Hospital Construction Fund and Shenzhen Basic Research Key Project (JCYJ20220818102811024); Lam Chung Nin Foundation for Systems Biomedicine
- Please upload the "Source Data of Primers" (currently in an archive) either directly as an "Expanded View Table" in DOCX or XLSX format, and reference it at least once in the methods section as "Table EV1"; or include the primer sequences in the provided "Reagents and Tools Table".
- Please upload all Figure Source Data in their native file format - i.e. images as JPG/PNG/PDF, numerical data as XLSX, ...; but avoid e.g. images pasted into spreadsheets. Please combine all Appendix Figure source data in one single archive file; whereas source data for main figures should be in one archive per each main figure.

I am therefore returning the manuscript to you for a final round of minor revision, to allow you to make the requested presentational and editorial modifications, and upload the revised files. Once we will have received them, we should hopefully be ready to swiftly proceed with formal acceptance and production of the manuscript.

Yours sincerely,

Hartmut Vodermaier

Hartmut Vodermaier, PhD
Senior Editor, The EMBO Journal

*** PLEASE NOTE: All revised manuscripts are subject to initial checks for completeness and adherence to our formatting guidelines. Revisions may be returned to the authors and delayed in their editorial re-evaluation if they fail to comply to the following requirements (see also our Guide to Authors for further information):

9) Digital image enhancement is acceptable practice, as long as it accurately represents the original data and conforms to community standards. If a figure has been subjected to significant electronic manipulation, this must be clearly noted in the figure legend and/or the 'Materials and Methods' section. The editors reserve the right to request original versions of figures and the original images that were used to assemble the figure. Finally, we generally encourage uploading of numerical as well as gel/blot image source data; for details see: embopress.org/page/journal/14602075/authorguide#sourcedata

At EMBO Press, we ask authors to provide source data for the main manuscript figures. Our source data coordinator will contact you to discuss which figure panels we would need source data for and will also provide you with helpful tips on how to upload and organize the files.

Further information is available in our Guide For Authors:

In the interest of ensuring the conceptual advance provided by the work, we recommend submitting a revision within 3 months (22nd Apr 2024). Please discuss the revision progress ahead of this time with the editor if you require more time to complete the revisions. Use the link below to submit your revision:

Link Not Available

Referee #1:

I would like to express my appreciation for the significant efforts you have put into revising your manuscript. It is evident that the manuscript has undergone considerable improvement. The addition of quantitative mutational analysis is particularly compelling and adds substantial weight to the claims presented in the paper. However, structural quality remains still mediocre but I appreciate the extra effort and believe the claims made are justified.

I do have some minor suggestions that could further enhance the clarity and overall quality of the manuscript. It would be beneficial to refine further the figures to improve their clarity. Additionally, ensuring consistency in terms of colors and arrangement in the figures would contribute to a more cohesive and professional presentation.

Lastly, I commend the inclusion of additional quantitative data in your analysis. However, I noticed that the number of digits used in some instances appears excessive especially in the tables. To enhance the precision and readability of your data, I recommend adjusting these numbers to reflect only significant digits.

Referee #3:

The revised study on CRL2FEM1B from Dai, Liang, and Wang "Dai et al." has greatly improved clarity with new experimental results to address reviewer concerns. The main findings are their CRL2FEM1B structures from single particle cryo-EM, including +/- Neddylation and investigating CRL2FEM1B substrates BEX2 and FNIP1. Overall CUL2 is a scaffold for several important substrates/adaptors that forms essential CRL2 based E3 ubiquitin ligases. The Dai et al. study provides functional and mutations analysis to understand how CRL2FEM1B functions as an E3 ligase and investigates the oligomerization of CRL2FEM1B. Their structures of asymmetric dimer and trimer are supported through biophysical methods and unreported. In general, CRL2 E3s are challenging structurally and this Dai et al. study advances our understanding of cullin-RING E3 ligases (CRLs). X-ray derived structures of FEM1B have been reported, but the combination of FEM1B on CRL2 makes their structures novel. In addition, their general approach and methodology is convincing to demonstrate CRL2FEM1B oligomers.

In response to myself, reviewer 3, I found the new changes greatly improved my understanding of their study. Overall, almost all my concerns were addressed, confusing sections now have clarity, and the figures are much easier to understand.

With the cryo-EM data the question is, what is the limit of each data set and has sufficient work been carried out to understand the limit?

I would accept this study with many of the minor revisions.

Minor points:

The addition of the binding data from Creoptix WAVE and thermal stability from nanoDSF are informative.

For the cryo-EM Table 1 (Appendix Table S1), this appears to be very narrow defocus range for all data sets.

Some of the first derivative melting curves for CRL2-FEM1B (Appendix Figure S11 A-F) suggest different domains or subunits of the CRL experience thermal unfolding at different temperatures i.e. multiple unfolding events. Is this notable?

In your "Reagents and Tools Table" under software, was the Mass Photometry (MP) software from Refeyn used (e.g. Refeyn: AcquireMP or DiscoverMP). Any software to note from Nanotemper or for the WAVE in this table?

Do you need to specify the version of Topaz (<https://github.com/tbepler/topaz> v0.2.5) or deepEMhancer (<https://github.com/rsanchezgarc/deepEMhancer>)?

As a reviewer I appreciated the link to google drive <https://drive.google.com/drive/folders/1qvyaAADRJZWGMn0JAS0EDnNyx6iNyspA?us%20638%20p=sharing> the owner was hidden and I could access the volumes (.mrc) from any web browser without logging into a google account - making the whole system anonymous from end-to-end.

All of the volumes (.mrc) in Google drive seem to have *_map_deepem.mrc and these would be presumably output from deepEMhancer processing of an unsharpened map. In some cases, the unsharpened maps could also be helpful (e.g. if someone wanted to carry out automated model building).

Each of the three maps appears to have areas that are not modelled with sometimes large volumes that could be another subunit.

You mention "Dr. Oliver B. Clarke (Columbia)" who is regarded as an expert contributor to Cryosparc users (<https://guide.cryosparc.com/processing-data/tutorials-and-case-studies/case-study-exploratory-data-processing-by-oliver-clarke>) / discuss user "olibclarke." I believe he would advise you to include or at least try different processing approaches that go beyond your current workflow.

For instance, following the schemes from Appendix Figure S2, S3, S4 - the multiclass ab-initio leads to several rounds of heterogeneous refinement without the 3D classification job or 3DVA. Use of a refinement mask may not work (small size, not enough signals), but particle subtraction with a refinement mask in local refine could help resolve areas in the larger CRL2FEM1B structures.

The authors point that the areas of the interfaces are resolved is valid and the local resolution across CRL2FEM1B does vary greatly as it is a flexible system. However, many in the cryo-EM field would inspect the map-to-model. Then after examining the processing workflow, they would wonder if more analysis could have benefited these structures.

Some questions would be:

1. Was binning or "Fourier crop" to box size applied in Cryosparc, if so at what steps ?
2. What was the box size (in pixels) and when were particles re-extracted at the full size ?
3. Could Global CTF refinement, Local CTF refinement, or reference based motion correction improve anything ?

N8_CRL2FEM1B_FNIP1_map_deepem.mrc has some strange artefacts, maybe cleaning strength in deepEMhancer ?

Point-by-Point Response

We appreciate the reviewers and editor for valuable comments provided and considering our article for publication. Our point-by-point response is the following:

Editorial issues:

1. Please make sure to reference Appendix Table S1 (cryo-EM data collection) at least once in the main text.

Response:

Thank you for your advice. We have mentioned Appendix Table S1 as suggested in the main text. “Using conventional single particle analysis cryo-EM method, we solved the structure of purified recombinant CRL2^{FEM1B} to 4.08 Å resolution (Appendix Table S1, Appendix Figure S2A and B).” “To investigate the structural basis of ubiquitylation of BEX2 by neddylated CRL2^{FEM1B}, we determined the cryo-EM structure of N8-CRL2^{FEM1B} complexed with MBP-BEX2 (Appendix Table S1, Appendix Figure S3A and B and Appendix Figure S5A).” “To determine whether the ubiquitylation strategy of this asymmetric N8-CRL2^{FEM1B} dimer is applicable to other substrates, we obtained the stable N8-CRL2^{FEM1B}-FNIP1-FLCN complex and its cryo-EM structure (Appendix Table S1, Appendix Figure S4A and B and Appendix Figure S5A).”

2. Please double-check all citations in the reference list, as some of them appear to be still incomplete (lacking page/locator numbers).

Response:

Thank you for reminding us. We have revised the reference list as suggested.

3. Please rename the Conflict of Interest section into "Disclosure and Competing Interests Statement", in accordance with our updated Guide to Authors (<https://www.embopress.org/competing-interests>)

Response:

We have renamed the section into “Disclosure and Competing Interests Statement” accordingly.

4. As we are switching from a free-text author contribution statement towards a more formal statement based on Contributor Role Taxonomy (CRediT) terms, please remove the present Author Contribution section and instead specify each author's contribution(s) directly in the Author Information page of our submission system during upload of the final manuscript. See <https://casrai.org/credit/> for more information.

Response:

We have removed the author contribution section and input into the submission system as suggested.

5. Please rename the "Data and materials availability" section into simply "Data Availability", and delete its first sentence. Please also remove the included reviewer access information at this point, and ensure that data become publicly accessible upon acceptance.

Response:

We have renamed the “Data Availability” section as suggested.

6. Please double-check to make sure to all relevant funding information in the manuscript is congruent with the info entered into our submission system. Currently missing in the submission system are: The Shenzhen High-level Hospital Construction Fund and Shenzhen Basic Research Key Project (JCYJ20220818102811024); Lam Chung Nin Foundation for Systems Biomedicine

Response:

We have input the missing funding information into the submission system as suggested.

7. Please upload the "Source Data of Primers" (currently in an archive) either directly as an "Expanded View Table" in DOCX or XLSX format, and reference it at least once in the methods

section as "Table EV1"; or include the primer sequences in the provided "Reagents and Tools Table".

Response:

We have included the primer sequences in the "Reagents and Tools Table" as suggested.

8. Please upload all Figure Source Data in their native file format - i.e. images as JPG/PNG/PDF, numerical data as XLSX, ...; but avoid e.g. images pasted into spreadsheets. Please combine all Appendix Figure source data in one single archive file; whereas source data for main figures should be in one archive per each main figure.

Response:

All figure source data are prepared as suggested including one archive file for appendix figure source data and archive files for each main figure.

Referee #1:

1. I would like to express my appreciation for the significant efforts you have put into revising your manuscript. It is evident that the manuscript has undergone considerable improvement. The addition of quantitative mutational analysis is particularly compelling and adds substantial weight to the claims presented in the paper. However, structural quality remains still mediocre but I appreciate the extra effort and believe the claims made are justified.

Response:

Thank you so much for your kind comments. We have to admit that it's a challenging task for us to further improve the quality of our structures and we appreciate your recognition of our efforts.

2. I do have some minor suggestions that could further enhance the clarity and overall quality of the manuscript. It would be beneficial to refine further the figures to improve their clarity. Additionally, ensuring consistency in terms of colors and arrangement in the figures would contribute to a more cohesive and professional presentation.

Response:

Thank you for your advice. We have revised Figure 1C-E, Figure 3A-F, Figure 8A-E and Appendix Figure S6A-D to ensure the consistency of the colors. As for Figure 6, we kept the original coloring style. Because we believe that it will show N8-CRL2^{FEM1B}-BEX2 complex has a similar conformation comparing to Protomer1 and protomer 2 in CRL2^{FEM1B} complex but it's a different state.

Figure 1 C-D

Figure 3A-F

Figure 8A-E

Figure S6A-D

3. Lastly, I commend the inclusion of additional quantitative data in your analysis. However, I noticed that the number of digits used in some instances appears excessive especially in the tables. To enhance the precision and readability of your data, I recommend adjusting these numbers to reflect only significant digits.

Response:

Thank you for your recognition. We have removed some unrelated digits in the tables such as the molecular weight of excessive FNIP1/FCLN complex in Table 3 and late-stage T_m of $CRL2^{FEM1B}$ complexes in Table 1 to enhance the readability.

Table 1	
The effect of mutations on CRL2FEM1B and MBP-FCB protein stability (n=3)	
Samples	T_m (°C)
CRL2FEM1B-WT	56.28±0.7
CRL2FEM1B-DEL	47.57±0.25
CRL2FEM1B-F549R	50.51±0.03
CRL2FEM1B-F549S	50.4±0.13
CRL2FEM1B-F549T	53.78±0.03
CRL2FEM1B-Y275R/L278R	48.91±0.05
MBP-FCB-WT	46.78±0.01
MBP-FCB-DEL	47.23±0.05
MBP-FCB-F549R	46.58±0.02
MBP-FCB-F549S	46.69±0.02
MBP-FCB-F549T	46.7±0.02
MBP-FCB-Y275R/L278R	47.69±0.02

Table 1

Table 3			
Molecular weights of wild-type and mutants of (N8-)CRL2FEM1B and E3-substrate complexes measured by mass photometry experiments			
Sample	Molecular Weight (kDa)		
CRL2FEM1B-WT	191±1	414±31	576±3
CRL2FEM1B-DEL	191±9		
CRL2FEM1B-F549R	189±3	386±13	
CRL2FEM1B-F549T	186±0	384±21	
CRL2FEM1B-F549S	187±3	375±11	
CRL2FEM1B-Y275RL278R	188±11		
N8-CRL2FEM1B-WT	197±5	402±13	
N8-CRL2FEM1B-DEL	197±27		
N8-CRL2FEM1B-F549R	203±28		
N8-CRL2FEM1B-F549T	190±18	395±11	
N8-CRL2FEM1B-F549S	192±17		
N8-CRL2FEM1B-Y275RL278R	183±7		
N8-CRL2FEM1B-WT-BEX2		517±36	
N8-CRL2FEM1B-WT-FNIP1/FLCN		380±12	

Table 3

Referee #3:

1. In response to myself, reviewer 3, I found the new changes greatly improved my understanding of their study. Overall, almost all my concerns were addressed, confusing sections now have clarity, and the figures are much easier to understand.

With the cryo-EM data the question is, what is the limit of each data set and has sufficient work been carried out to understand the limit?

Response:

Thank you for your recognition of our efforts. For the data set of unneddylated CRL2^{FEM1B} complex, we lost most of the micrographs at the stage of “manually curate exposures” (1855 out of 5400) with a cutoff of CTF fit resolution at 5 Å. We believe that we still don’t have sufficient number of particles during high-resolution reconstruction even after TOPAZ picking and seed based heterogenous refinement. As for the data set of neddylated CRL2^{FEM1B}-BEX2 complex, we believe that the heterogeneity of sample itself and the flexibility of catalytic domain and

substrate recognition domain limited the final resolution of reconstruction. For the data set of neddylated CRL2^{FEM1B}-FNIP1/FLCN complex, we collected 15,565 micrographs aiming at resolving the structure of substrate-E3 binding domain. However, the unpredictable movement of substrate FNIP1/FLCN made this task nearly impossible. The binding site located on a flexible loop of FNIP1 making FNIP1/FLCN very dynamic relatively to FEM1B. Moreover, as predicted by alphafold, FNIP1 has quite few rigid domains. Meanwhile, neddylation destabilize the catalytic domain of CRL2^{FEM1B}, which limits us to achieve higher resolution reconstruction.

2. For the cryo-EM Table 1 (Appendix Table S1), this appears to be very narrow defocus range for all data sets.

Response:

The defocus range shown in the cryo-EM Table 1 is the parameter that we input into the EPU. Because we were using UltraAu Foil grids, the actual defocus range was larger than the parameter that we set.

3. Some of the first derivative melting curves for CRL2-FEM1B (Appendix Figure S11 A-F) suggest different domains or subunits of the CRL experience thermal unfolding at different temperatures i.e. multiple unfolding events. Is this notable?

Response:

Multiple T_m values do indicate multiple unfolding events. As we couldn't observe multiple unfolding events in MBP-FCB complexes and its mutations, we believe that those unfolding events might related to the states of CUL2 and RBX1 subunits in the CRL2^{FEM1B} complexes. It is indeed an interesting phenomenon that draws our attention in the future studies.

4. In your "Reagents and Tools Table" under software, was the Mass Photometry (MP) software from Refeyn used (e.g. Refeyn: AcquireMP or DiscoverMP). Any software to note from Nanotemper or for the WAVE in this table?

Response:

Thank you for your advice. We have added the information into the “Reagents and Tools Table” as suggested.

DiscoverMP software	Refeyn Ltd, https://refeyn.filecamp.com/
	UK 1
PR.ThermoControl	NanoTemper N/A
Creoptix WAVEcontrol software	Creoptix AG, N/A Switzerland

Reagents and Tools Table

5. Do you need to specify the version of Topaz ([https://github.com/tbepler/topaz v0.2.5](https://github.com/tbepler/topaz)) or deepEMhancer (<https://github.com/rsanchezgarc/deepEMhancer>) ?

Response:

Thank you for reminding us. We have added the version of TOPAZ and DeepEMhancer in the “Reagents and Tools Table” and main text as suggested.

TOPAZ 0.2.5a	(Bepler et al , https://github.com/tbepler/topaz 2019)
DeepEMhancer	(Sanchez-Garcia et al , https://github.com/rsanchezgarc/deepEMhancer 2021)

Reagents and Tools Table

“Best particles were then used as template for TOPAZ (v0.2.5a) particle picking (Bepler *et al*, 2019).”

6. All of the volumes (.mrc) in Google drive seem to have *_map_deepem.mrc and these would be presumably output from deepEMhancer processing of an unsharpened map. In some cases, the unsharpened maps could also be helpful (e.g. if someone wanted to carryout automated model building).

Response:

Thank you for your advice. We have uploaded the unsharpened maps to google drive as suggested. The link to google drive is shown below:

https://drive.google.com/drive/folders/1qvyaAADRJZWGMn0JAS0EDnNyx6iNyspA?usp=drive_link

7. Each of the three maps appears to have areas that are not modelled with sometimes large volumes that could be another subunit.

Response:

In the map of unnedylated CRL2^{FEM1B}, the large volume that was not modelled corresponds to protomer 3 of the trimeric complex. Because the local resolution of that volume was not good enough for model building, we fitted the model of protomer 2 into that density for interpretation. In the map of neddylated CRL2^{FEM1B}-BEX2 complex, the large volume corresponds to the catalytic domain of protomer 2 which includes RBX1, WHB domain and NEDD8. However, the local resolution was too low to build atomic models. In the map of neddylated CRL2^{FEM1B}-FNIP1/FLCN complex, the large volume corresponds to the FNIP1/FLCN complex which is highly flexible. We were not confident enough to fit models into that volume.

8. You mention "Dr. Oliver B. Clarke (Columbia)" who is regarded as an expert contributor to Cryosparc users (<https://guide.cryosparc.com/processing-data/tutorials-and-case-studies/case-study-exploratory-data-processing-by-oliver-clarke>) / discuss user "olibclarke." I believe he would advise you to include or at least try different processing approaches that go beyond your current workflow.

Response:

Dr. Clarke did provide advice during the early stage of this project and the build-up of our data-processing work station. In the later stage of this project, he provided same suggestions as yours, such as try TOPAZ picking, Resolve cryo-EM in Phenix, CTF refinement, local refinement and 3DVA as well. We are very grateful to both of your valuable suggestions.

9. For instance, following the schemes from Appendix Figure S2, S3, S4 - the multiclass ab-initio leads to several rounds of heterogeneous refinement without the 3D classification job or 3DVA. Use of a refinement mask may not work (small size, not enough signals), but particle subtraction with a refinement mask in local refine could help resolve areas in the larger CRL2^{FEM1B} structures.

Response:

Thank you for your suggestions. We tried 3DVA in all three data sets and the results are partially shown in Appendix Figure S10. As we can see, the density of protomer 3 in the trimeric CRL2^{FEM1B} remains weak in the output of 3DVA. Meanwhile, the local resolution of the N terminal of FEM1B and C terminal of CUL2 in protomer 2 of N8-CRL2^{FEM1B}-BEX2 complex are still low. We compared the outputs from local refinement and NU refinement which showed no significant difference in the quality of cryo-EM maps.

Fig. S10. 3D variability analysis of CRL2^{FEM1B} and N8-CRL2^{FEM1B}-BEX2 complexes.

(A) 3DVA analysis of trimeric CRL2^{FEM1B} cryo-EM data by cryoSPARC shows the flexibility of the substrate binding pocket of FEM1B. Broken circled region correspond to the density of FEM1B within trimeric CRL2^{FEM1B} complex.

(B) 3DVA analysis of N8-CRL2^{FEM1B}-BEX2 cryo-EM data by cryoSPARC shows the flexibility of the catalytic domain of protomer 2.

Appendix Figure S10

10. The authors point that the areas of the interfaces are resolved is valid and the local resolution across CRL2^{FEM1B} does vary greatly as it is a flexible system. However, many in the cryo-EM field would inspect the map-to-model. Then after examining the processing workflow, they would wonder if more analysis could have benefited these structures.

Response:

Thank you for your suggestions. We have measured the model-map Q-Score of all three structures in ChimeraX. And the results are shown below.

1. Trimeric CRL2FEM1B Model-map Q-Score:

Overall mean Q-Score: 0.33

However, the Q-score of side chains of FEM1B of protomer 1 and protomer 2 at the dimerization interface are around 0.6.

2. Dimeric N8-CRL2FEM1B-BEX2 Model-map Q-Score:

Overall mean Q-Score: 0.44

However, the Q-score of side chains of FEM1B of protomer 1 and protomer 2 at the dimerization interface are around 0.6.

3.N8-CRL2FEM1B-FNIP1/FLCN Model-map Q-Score:

Overall mean Q-Score:0.35, the Q-score of the residues of FEM1B is shown below.

11. Was binning or "Fourier crop" to box size applied in Cryosparc, if so at what steps? What was the box size (in pixels) and when were particles re-extracted at the full size?

Response:

For all three data sets, we collected movies in super-resolution mode and applied output fourier cropping factor $\frac{1}{2}$ during motion correction. For dimeric CRL2^{FEM1B} complex, we extracted particles with a box size of 480 pixels without applying any "Fourier crop" during the whole

process as there were not a lot of particles. For N8-CRL2^{FEM1B}-BEX2 complex, we extracted particles with a box size of 512 pixels and binned to 128 pixels after template picking. After training TOPAZ picking with best particles from template picking, we extracted particles with a box size of 480 pixels at the full size. For N8-CRL2^{FEM1B}-FNIP1/FLCN complex, we extracted particles with a box size of 480 pixels and binned to 120 pixels after particle picking. After cleaning junk particles with 2D classification and seed based heterogenous refinements, we re-extracted clean particles with a box size of 480 pixels at the full size to the end of data processing.

12. Could Global CTF refinement, Local CTF refinement, or reference based motion correction improve anything?

Response:

Thank you so much for your advice. We have tried global CTF refinement, local CTF refinement and their combination during our data-processing. However, the resolution of cryo-EM maps was not improved in these cases. Reference based motion correction is bundled in the latest version of CryoSPARC v4.4 which we are definitely planning to try.

13. N8_CRL2FEM1B_FNIP1_map_deepem.mrc has some strange artefacts, maybe cleaning strength in deepEMhancer?

Response:

The artefacts are caused by the bad choice of 'tightTarget' model of DeepEMhancer. It could be avoided by choosing 'wideTarget' model of DeepEMhancer. New map is also uploaded onto the google driver mentioned above. The figure is shown below.

Sharpened cryo-EM map of N8-CRL2^{FEM1B}-FNIP1/FLCN complex

Prof. Yuxin Yin
Peking University Health Science Center
Institute of Systems Biomedicine
38 Xue Yuan Road
Beijing, Beijing 100191
China

29th Jan 2024

Re: EMBOJ-2023-115372R1
Structural Insights into the Ubiquitylation Strategy of the Oligomeric CRL2FEM1B E3 Ubiquitin Ligase

Dear Prof. Yin,

Thank you for submitting your final revised manuscript for our consideration. I am pleased to inform you that we have now accepted it for publication in The EMBO Journal.

Yours sincerely,

Hartmut Vodermaier
